# Short- and long-read metagenomics of urban and rural South African gut microbiomes reveal a transitional composition and undescribed taxa

Fiona B. Tamburini [1], Dylan Maghini[1], Ovokeraye H. Oduaran [2], Ryan Brewster[3], Michaella R. Hulley[2,4], Venesa Sahibdeen[4], Shane A. Norris [5,6], Stephen Tollman [7,8], Kathleen Kahn [7,8], Ryan G. Wagner[7,8], Alisha N. Wade[7], Floidy Wafawanaka[7], F. Xavier Gómez-Olivé [7,8], Rhian Twine[7], Zané Lombard[4], H3Africa AWI-Gen Collaborative Centre, Scott Hazelhurst [2,9,11✉] & Ami S. Bhatt [1,3,10,11✉]

Human gut microbiome research focuses on populations living in high-income countries and to a lesser extent, non-urban agriculturalist and hunter-gatherer societies. The scarcity of research between these extremes limits our understanding of how the gut microbiota relates to health and disease in the majority of the world's population. Here, we evaluate gut microbiome composition in transitioning South African populations using short- and long-read sequencing. We analyze stool from adult females living in rural Bushbuckridge ($n = 118$) or urban Soweto ($n = 51$) and find that these microbiomes are taxonomically intermediate between those of individuals living in high-income countries and traditional communities. We demonstrate that reference collections are incomplete for characterizing microbiomes of individuals living outside high-income countries, yielding artificially low beta diversity measurements, and generate complete genomes of undescribed taxa, including *Treponema*, Lentisphaerae, and *Succinatimonas*. Our results suggest that the gut microbiome of South Africans does not conform to a simple "western-nonwestern" axis and contains undescribed microbial diversity.

[1] Department of Genetics, Stanford University, Stanford, CA, USA. [2] Sydney Brenner Institute for Molecular Bioscience, University of the Witwatersrand, Johannesburg, South Africa. [3] School of Medicine, Stanford University, Stanford, CA, USA. [4] Division of Human Genetics, School of Pathology, Faculty of Health Sciences, National Health Laboratory Service & University of the Witwatersrand, Johannesburg, South Africa. [5] SAMRC Developmental Pathways for Health Research Unit, Department of Paediatrics, University of the Witwatersrand, Johannesburg, South Africa. [6] School of Human Development and Health, University of Southampton, Southampton, UK. [7] MRC/Wits Rural Public Health and Health Transitions Research Unit (Agincourt), School of Public Health, Faculty of Health Sciences, University of the Witwatersrand, Johannesburg, South Africa. [8] INDEPTH Network, East Legon, Accra, Ghana. [9] School of Electrical and Information Engineering, University of the Witwatersrand, Johannesburg, South Africa. [10] Department of Medicine (Hematology, Blood and Marrow Transplantation), Stanford University, Stanford, CA, USA. [11] These authors contributed equally: Scott Hazelhurst, Ami S. Bhatt. ✉email: Scott.Hazelhurst@wits.ac.za; asbhatt@stanford.edu

Comprehensive characterization of the full diversity of the healthy human gut microbiota is essential to contextualize studies of the microbiome related to diet, lifestyle, and disease. To date, substantial resources have been invested in describing the microbiome of individuals living in the global industrialized "west" (United States, northern and western Europe; also sometimes referred to as the "Global North"), including efforts by large consortia such as the Human Microbiome Project[1] and MetaHIT[2]. Though these projects have yielded valuable descriptions of human gut microbial ecology, they survey only a small portion of the world's citizens at the extreme of industrialized, urbanized lifestyle. It is unclear to what extent these results are generalizable to nonwestern and non-industrialized populations across the globe.

At the other extreme, a smaller number of studies have characterized the gut microbiome composition of individuals practicing traditional lifestyles[3,4], including communities in Venezuela and Malawi[5], hunter-gatherer communities in Tanzania[6–9], non-industrialized populations in Tanzania and Botswana[10], and agriculturalists in Peru[11] and remote Madagascar[12]. However, these cohorts are not representative of how most of the world lives either. Many of the world's communities lead lifestyles between the extremes of an urbanized, industrialized and relatively high-income lifestyle and traditional subsistence practices. It is a scientific and ethical imperative to include these diverse populations in biomedical research, yet dismayingly many of these intermediate groups are underrepresented in or absent from the published microbiome literature.

This major gap in our knowledge of the human gut microbiome leaves the biomedical research community ill-poised to relate microbiome composition to human health and disease across the breadth of the world's population. Worldwide, many communities are currently undergoing a transition of diet and lifestyle, characterized by increased access to processed foods, diets rich in animal fats and simple carbohydrates, and more sedentary lifestyles[13]. This has corresponded with an epidemiological transition in which the burden of disease is shifting from predominantly infectious diseases to an increasing incidence of noncommunicable diseases (NCDs) like obesity and diabetes[14]. The microbiome has been implicated in various NCDs[15–17] and may mediate the efficacy of medical interventions including vaccines[18,19], but we cannot evaluate the generalizability of these findings without establishing baseline microbiome characteristics of communities that practice diverse lifestyles and by extension, harbor diverse microbiota. These understudied populations, which are more representative of the majority of the world's population, offer a unique opportunity to examine the relationship between lifestyle (including diet), disease, and gut microbiome composition, and to discover novel microbial genomic content that may associate with or drive disease biology.

Some previous studies have probed the relationship between lifestyle and microbiome composition in transitional communities[3,20–22]. However, substantial gaps remain in our description of the microbiome in these populations. In particular, knowledge of the gut microbiota within the African continent is sparse. Of the 64 studies surveying the gut microbiome of individuals living within Africa as of January 2021 (Supplementary Data 1), only 25 of the 54 countries (46%) on the continent are represented. Of these studies, 34 of 64 (53%) have focused entirely on children or infants, whose disease risk profile and gut microbiome composition can vary considerably from adults[5,23]. Additionally, 52 of 64 (81%) of studies of the gut microbiome in Africans employed 16S ribosomal RNA (rRNA) gene sequencing or quantitative polymerase chain reaction (qPCR), techniques which amplify only a small portion of the genome and therefore lack genomic resolution to describe species or strains that may share a 16S rRNA sequence but differ in gene content or genome structure. To our knowledge, only nine published studies to date have used shotgun metagenomics to describe the gut microbiome of adults living in Africa. Eight of these studies described the bacterial microbiome[6,7,12,24–28], while one exclusively described the viral metagenome[29].

To address this major knowledge gap, we designed and performed a research study applying short- and long-read DNA sequencing to study the gut microbiomes of South African individuals for whom 16S rRNA gene sequence data has recently been reported[30]. South Africa is a prime example of a country undergoing rapid lifestyle and epidemiological transition. With the exception of the HIV/AIDS epidemic in the mid-1990s to the mid-2000s, over the past three decades South Africa has experienced a steadily decreasing mortality rate from infectious disease and an increase in NCD[31,32]. Concomitantly, increasingly sedentary lifestyles and changes in dietary habits, including access to calorie-dense processed foods, contribute to a higher prevalence of obesity in many regions of South Africa[32], a trend which disproportionately affects women[33,34].

This study presents the largest shotgun metagenomic dataset of African adults in the published literature to date. In this work, we describe microbial community-scale similarities between urban and rural communities in South Africa, as well as distinct hallmark taxa that distinguish each community. Additionally, we place South African populations in context with microbiome data from other populations from countries around the world, revealing the transitional nature of gut microbiome composition in the South African cohorts. We demonstrate that metagenomic assembly of short reads yields previously undescribed strain and species draft genomes. Finally, we apply Oxford Nanopore long-read sequencing to samples from the rural cohort and generate complete and near-complete genomes. These include genomes of species that are exclusive to, or more prevalent in, traditional populations, including Treponema and Prevotella species. As long-read sequencing enables more uniform coverage of AT-rich regions compared to short-read sequencing with transposase-based library preparation, we also generate complete metagenome-assembled AT-rich genomes from less well-described gut microbes including species in the phylum Melainabacteria, the class Mollicutes, and the genus Mycoplasma.

Taken together, the results herein offer a more detailed description of gut microbiome composition in understudied transitioning populations, and present complete and contiguous reference genomes that will enable further studies of gut microbiota in nonwestern populations. Importantly, this study was developed with an ethical commitment to engaging both rural and urban community members to ensure that the research was conducted equitably (additional details in Supplementary Information). This work underscores the critical need to broaden the scope of human gut microbiome research and include understudied, nonwestern populations to improve the relevance and accuracy of microbiome discoveries to broader populations.

## Results

**Cohorts and sample collection.** We enrolled 190 women aged between 40 and 72, living in rural villages in the Bushbuckridge Municipality (24.82°S, 31.26°E, $n = 132$) and urban Soweto, Johannesburg (26.25°S, 27.85°E, $n = 58$) and collected a one-time stool sample, as well as point of care blood glucose and blood pressure measurements and a rapid HIV test. As HIV status and exposure to antiretroviral medications can alter the microbiome and potentially confound analyses, only samples from HIV-negative individuals were analyzed further ($n = 118$ Bushbuckridge, $n = 51$ Soweto). Participants spanned a range of body mass

**Table 1 Participant characteristics.**

| Measurement | Bushbuckridge | Soweto |
|---|---|---|
| Age | 55.5 ± 7.8 (43–72) | 54.1 ± 5.9 (43–64) |
| Body mass index (BMI)[a] | 32.4 ± 8.0 (21.2–59.0) | 36.0 ± 9.3 (20.4–58.6) |
| Systolic blood pressure[b] | 137.0 ± 18.5 (103.5–186.5) | 135 ± 22.5 (96.0–193.0) |
| Diastolic blood pressure[b] | 84.0 ± 12.5 (52.5–119.0) | 89.9 ± 14.4 (58.0–119.0) |

[a]One Bushbuckridge participant's BMI measurement was excluded as the recorded value was too low to be physiologically possible and deemed to have been recorded in error. We could not validate the correct BMI for this participant and thus have omitted them from the BMI summary statistics.
[b]A second participant from Bushbuckridge had missing blood pressure measurements and is not included in blood pressure summary statistics.

index (BMI) from healthy to overweight; the most common comorbidity reported was hypertension, and many patients reported taking anti-hypertensive medication (18 of 118 (15%) in Bushbuckridge, 15 of 51 (29%) in Soweto) (Table 1 and Supplementary Table 1). Additional medications are summarized in Supplementary Table 1. We extracted DNA from each stool sample and conducted 150 base pair (bp) paired-end sequencing on the Illumina HiSeq 4000 platform. A median of 34.6 million (M) raw reads were generated per sample (range 11.4–100 M), and a median of 14.9 M reads (range 4.2–33.3 M) resulted after preprocessing including de-duplication, trimming, and human read removal (Supplementary Data 2).

**Gut microbial composition**. We taxonomically classified sequencing reads against a comprehensive custom reference database containing all microbial genomes in RefSeq and GenBank at "scaffold" quality or better as of January 2020 (177,626 genomes total). Concordant with observations from 16S rRNA gene sequencing of the same samples[30], we find that *Prevotella*, *Bacteroides*, and *Faecalibacterium* are the most abundant genera in most individuals across both study sites (Fig. 1a and Supplementary Fig. 1, Supplementary Data 3; species-level classifications in Supplementary Data 4). Additionally, in many individuals we observe taxa that are uncommon in western microbiomes, including members of the VANISH (Volatile and/or Associated Negatively with Industrialized Societies of Humans) taxa (families Prevotellaceae, Succinovibrionaceae, and Spirochaetaceae) such as *Prevotella, Treponema*, and *Succinatimonas*, which are higher in relative abundance in communities practicing traditional lifestyles compared to western industrialized populations[8,35] (Fig. 1b and Supplementary Data Files 3 and 4). The mean relative abundance of each VANISH genus is higher in Bushbuckridge than Soweto, though the difference is not statistically significant for *Paraprevotella* or *Sediminispirochaeta* (Fig. 1b, two-sided Wilcoxon rank-sum test). Within the Bushbuckridge cohort, we observe a bimodal distribution of the genera *Succinatimonas*, *Succinivibrio*, and *Treponema* (Supplementary Fig. 2a). While we do not identify any clinical or demographic features that associate with this distribution, we observe that VANISH taxa are weakly positively correlated with one another in metagenomes from both Bushbuckridge and Soweto (Supplementary Fig. 2b, c).

Intriguingly, we observe that an increased proportion of reads aligned to the human genome during preprocessing in samples from Soweto compared to Bushbuckridge (Supplementary Fig. 3, two-sided Wilcoxon rank-sum test $p = 1.66e-12$). This could potentially indicate higher inflammation and immune cell content or sloughing of intestinal epithelial cells in the urban Soweto cohort compared to rural Bushbuckridge.

**Rural and urban microbiomes cluster distinctly in MDS**. We hypothesized that lifestyle differences of participants residing in rural Bushbuckridge vs. urban Soweto might be associated with demonstrable differences in gut microbiome composition.

Bushbuckridge and Soweto differ markedly in their population density (53 and 6357 persons per $km^2$ respectively as of the 2011 census) as well as in lifestyle variables including the prevalence of flush toilets (6.8 vs. 91.6% of dwellings) and piped water (11.9 vs. 55% of dwellings) (additional site demographic information in Supplementary Table 2)[36]. Soweto is highly urbanized and has been so for several decades, while Bushbuckridge is classified as a rural community, although it is undergoing rapid epidemiological transition[37,38]. Bushbuckridge also has circular rural/urban migrancy typified by some (mostly male) members of a rural community working and living for extended periods in urban areas, while keeping their permanent rural home[39]. Although our participants all live in Bushbuckridge, this migrancy in the community contributes to making the boundary between rural and urban lifestyles more fluid. Comparing the two study populations at the community level, we find that samples from the two sites have distinct centroids (PERMANOVA $p < 0.001$, $R^2 = 0.037$) but overlap (Fig. 2a), though we note that the dispersion of the Soweto samples is greater than that of the Bushbuckridge samples (PERMDISP2 $p < 0.001$). Across the study population we observe a gradient of *Bacteroides* and *Prevotella* relative abundance (Supplementary Fig. 4). This may be the result of differences in diet across the study population at both sites, as *Bacteroides* has been proposed as a biomarker of westernized lifestyles while *Prevotella* has been proposed as a biomarker of nonwestern lifestyles[5,40,41].

To determine if medication usage was associated with gut microbiome composition, we included each participant's self-reported concomitant medications (summarized in Supplementary Table 1) to re-visualize the microbiome composition of samples in MDS by class of medication (Supplementary Fig. 5a, b). We find that self-reported medication is not significantly correlated with community composition in this cohort after multiple hypothesis correction (PERMANOVA FDR $q > 0.05$, Supplementary Fig. 5c), though two drug classes are nominally significant before controlling the false discovery rate: proton pump inhibitors (PPIs) ($p = 0.036$) and anti-hyperglycemics ($p = 0.041$). We note that both drug classes have previously been found to associate with changes in gut microbiome composition[42–44]; as only two participants self-report taking PPIs at the time of sampling, additional data are required to evaluate whether PPIs associate with microbiome composition in these South African populations.

**Rural and urban microbiomes differ in Shannon diversity and species composition**. Gut microbiome alpha diversity of individuals living traditional lifestyles has been reported to be higher than those living western lifestyles[9,11,40]. In keeping with this general trend, we find that alpha diversity (Shannon) is significantly higher in individuals living in rural Bushbuckridge vs. urban Soweto (Fig. 2b; two-sided Wilcoxon rank-sum test, $p = 0.042$). Using DESeq2 to identify microbial genera that are differentially abundant across study sites, we find that genera including *Bacteroides*, *Bifidobacterium*, and *Streptococcus* are

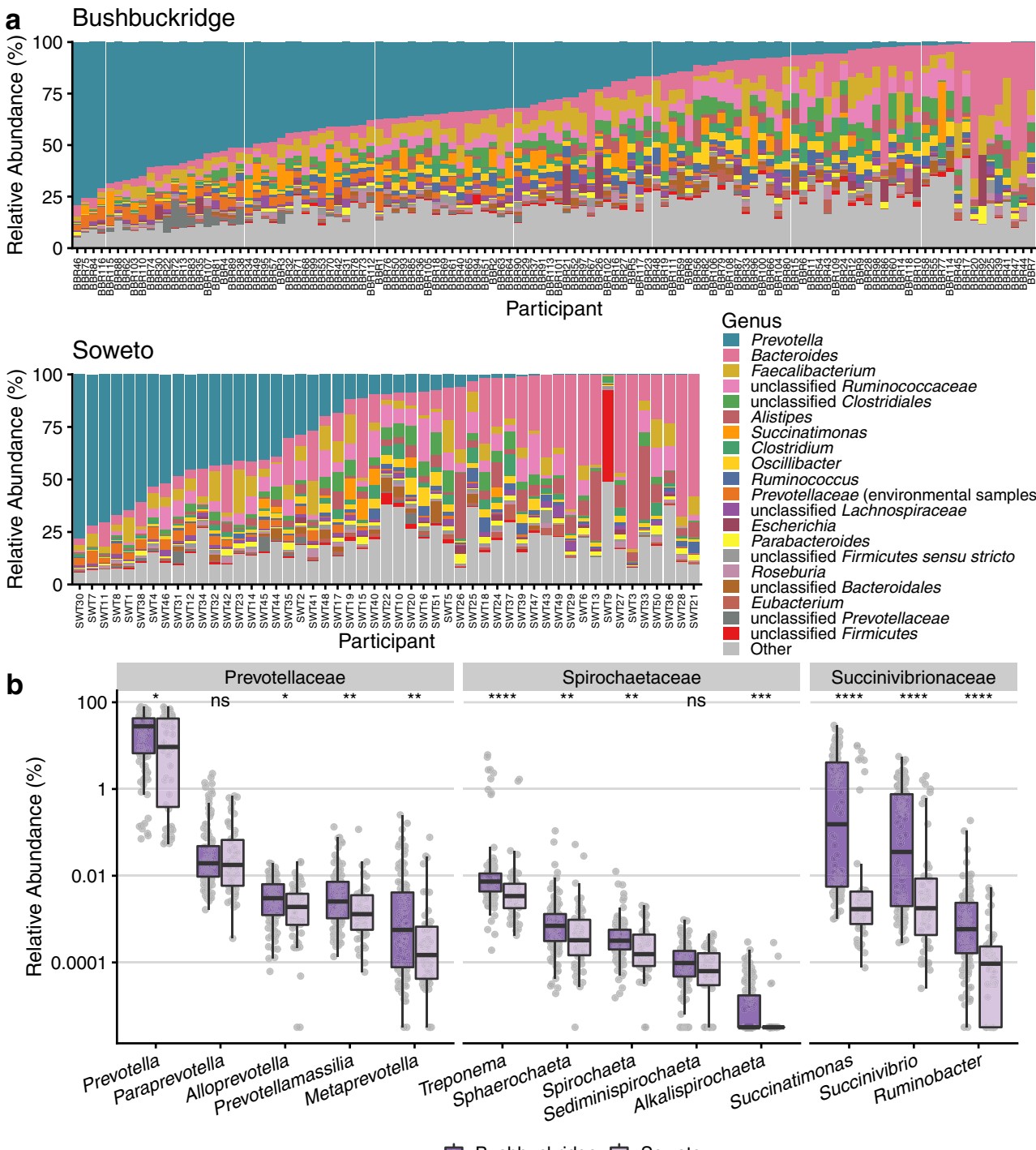

**Fig. 1 Taxonomic composition of South African study participant microbiota.** Sequence data were taxonomically classified using Kraken 2 with a database containing all genomes in RefSeq and GenBank of "scaffold" quality or better as of January 2020. **a** Top 20 genera by mean relative abundance for samples from participants in Bushbuckridge and Soweto, sorted by decreasing *Prevotella* abundance. *Prevotella*, *Bacteroides*, and *Faecalibacterium* are the most prevalent genera across both study sites. **b** Relative abundance of VANISH genera by study site, grouped by family (*n* = 118 Bushbuckridge, *n* = 51 Soweto). A pseudocount of 1 read was added to each sample prior to relative abundance normalization in order to plot on a log scale, as the abundance of some genera in some samples is zero. Relative abundance values of most VANISH genera are higher on average in participants from Bushbuckridge than Soweto (two-sided Wilcoxon rank-sum test, significance values denoted as follows: *$p$ < 0.05, **$p$ < 0.01, ***$p$ < 0.001, ****$p$ < 0.0001, (ns) not significant). Exact $p$ values from left to right: 3.91e−2, 3.28e−1, 1.60e−2, 4.55e−3, 6.64e−3, 1.93e−5, 9.20e−3, 7.29e−3, 6.93e−2, 6.87e−4, 1.64e−11, 7.66e−6, 1.02e−7. Box plot lower and upper hinges correspond to the first and third quartiles, upper and lower whiskers represent the highest and lowest values within 1.5 times the interquartile range, and the horizontal line represents the median.

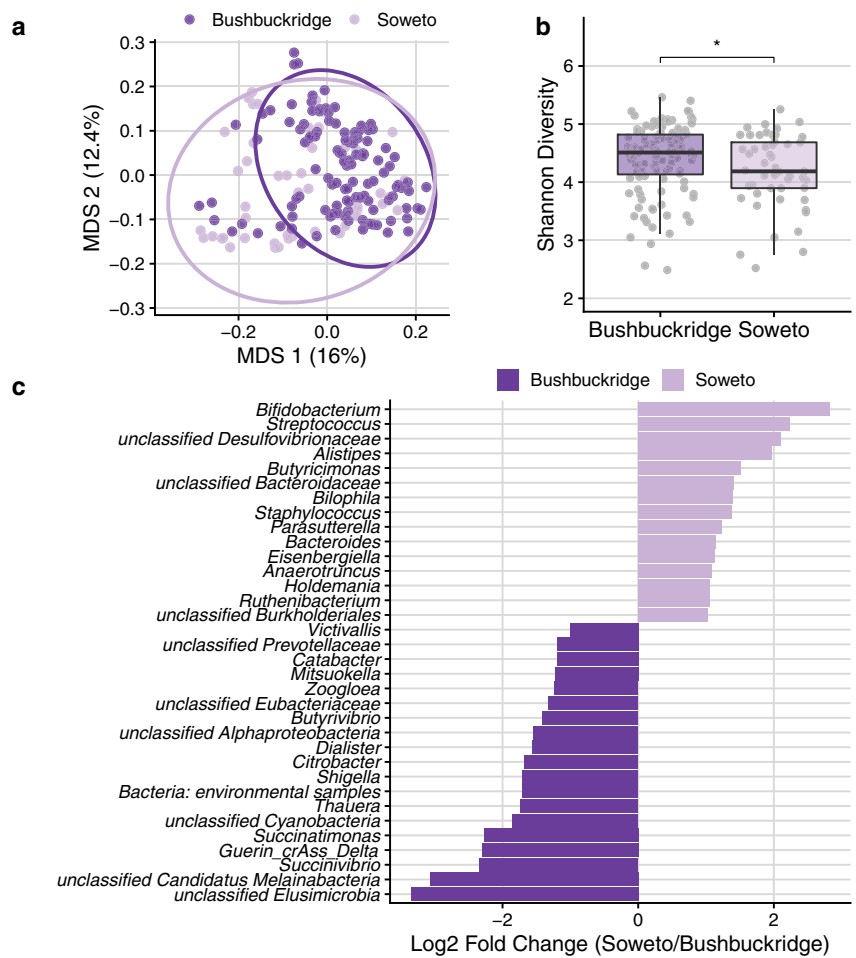

**Fig. 2 Comparison of Bushbuckridge and Soweto microbiomes. a** Multidimensional scaling (MDS) of pairwise Bray–Curtis distance between samples (rarefied to 1.44 M counts per sample to control for read depth and cumulative sum scaling normalized). Soweto samples have greater dispersion than Bushbuckridge (PERMDISP2 $p < 0.001$). **b** Shannon diversity calculated on rarefied species-level taxonomic classifications for each sample (participant $n = 118$ Bushbuckridge, $n = 51$ Soweto). Samples from Bushbuckridge are higher in alpha diversity than samples from Soweto (two-sided Wilcoxon rank-sum test, $p = 0.042$). For box plots, lower and upper hinges correspond to the first and third quartiles, upper and lower box plot whiskers represent the highest and lowest values within 1.5 times the interquartile range, and the horizontal line represents the median. **c** DESeq2 identifies microbial genera that are differentially abundant in rural Bushbuckridge compared to the urban Soweto cohort. Features with log2 fold change greater than one are plotted (full results in Supplementary Data 5).

more abundant in individuals living in Soweto (Fig. 2c and Supplementary Data 5, species shown in Supplementary Fig. 6). Interestingly, we find microbial genera enriched in gut microbiomes of individuals living in Bushbuckridge that are common to both the environment and the gut, including *Streptomyces* and *Paenibacillus* (Supplementary Data 5). Typically a soil-associated organism, *Streptomyces* encode a variety of biosynthetic gene clusters and can produce numerous immunomodulatory and anti-inflammatory compounds such as rapamycin and tacrolimus, and it has been suggested that decreased exposure to *Streptomyces* is associated with increased incidence of inflammatory disease and colon cancer in western populations[45]. In addition, we find enrichment of genera in Bushbuckridge that have been previously associated with nonwestern microbiomes including *Succinatimonas*, a relatively poorly-described bacterial genus with only one type species, and unclassified species of the phylum Elusimicrobia, which has been detected in the gut microbiome of rural Malagasy[12]. Additionally, Bushbuckridge samples are enriched for Cyanobacteria as well as Candidatus Melainabacter, a phylum closely related to Cyanobacteria that in limited studies has been described to inhabit the human gut[46,47].

In terms of the non-bacterial microbiome, we identify the bacteriophage crAssphage and related crAss-like phages[48], which have recently been described as prevalent constituents of the gut microbiome globally[49], in 32 of 51 participants (63%) in Soweto and 88 of 118 (75%) in Bushbuckridge (difference in prevalence between cohorts not significant, Fisher's exact test $p = 0.14$) using 650 sequencing reads or roughly 1X coverage of the 97 kilobase (kb) genome as a threshold for binary categorization of crAss-like phage presence or absence. Prototypical crAssphage has been hypothesized to infect *Bacteroides* species and a crAss-like phage has been demonstrated to infect *Bacteroides intestinalis*. Though crAss-like phages do not differ between cohorts in terms of prevalence (presence/absence), we observe that crAssphage clade Delta from Guerin et al.[48] is enriched in relative abundance in the gut microbiome of individuals living in Bushbuckridge compared to Soweto, supporting previous observations of geographic patterns of crAssphage clades (Fig. 2c)[49].

Our custom reference database of GenBank genomes paired with the Kraken 2 classifier optimizes for sensitivity; thus, this approach was selected as the initial tool for classification of the sequencing data given the genomic novelty anticipated in this cohort. We note that broadly similar microbiome profiles are

obtained using MetaPhlAn3, a marker-gene based tool with high specificity, as well as classifications obtained using Kraken 2 and a publicly available build of the Genome Taxonomy Database (GTDB) release 95[50,51] (Supplementary Fig. 7a, b). Notably, we observe higher Shannon diversity with the GTDB compared to both MetaPhlAn3 and our custom database, likely due to the fact that clades containing a large amount of genomic diversity (e.g., *Escherichia coli*) are split into separate clades in the GTDB (Supplementary Fig. 7c).

**Differences in functional potential of the gut microbiome between populations.** Recognizing that functional annotations are likely biased toward well-studied organisms, we sought to identify differentially abundant functions in the gut microbiome of participants in Bushbuckridge and Soweto. We functionally profiled unassembled metagenomic reads to detect antibiotic resistance genes in these communities. Tetracycline resistance genes (*tetO, tetQ, tetW, tetX, tet32, tet40*) are broadly prevalent in both populations (Supplementary Fig. 8a) as is the CfxA6 beta-lactamase. We find that Soweto and Bushbuckridge differ in the distribution of relative abundance of 30 of 113 antibiotic resistance genes (27%) with a model coefficient greater than 0.5 (Supplementary Fig. 8b). Several multidrug efflux pump components and regulators (*mdtB, mdtC, mdtF, mdtG, mdtL, mdtP, CRP*) are enriched in participants in Bushbuckridge, whereas genes including *SAT-4*, which is a plasmid-encoded streptothricin resistance determinant, and *CblA-1*, which encodes a class A beta-lactamase, are enriched in Soweto participants (Supplementary Fig. 8b).

We additionally annotated MetaCyc pathway abundance using HUMAnN v3[52] (Supplementary Data 6). We find 68 MetaCyc pathways that are differentially abundant between Soweto and Bushbuckridge ($q < 0.05$) (Supplementary Fig. 9a). Some of these pathways correspond clearly to observed taxonomic differences between study sites, including enrichment of the *Bifidobacterium* shunt, a pathway for degradation of hexose sugars into short chain fatty acids[53], in Soweto. Other differentially abundant pathways include anaerobic degradation of 4-coumarate, a phenylpropanoid compound produced by plants and by catabolism of the amino acid tyrosine[54]. Additionally, the superpathway of phenylethylamine degradation is enriched in Bushbuckridge. Intriguingly, phenylethylamine is a central nervous system stimulant in humans and increased abundance of phenylethylamine has been observed in Crohn's disease patients[55]. Finally, the peptidoglycan biosynthesis V pathway, involved in microbial resistance to beta-lactam antibiotics, is enriched in Soweto, consistent with results from antibiotic resistome profiling.

In general, HUMAnN was only able to ascribe functions to taxonomy for a few well-studied genera including *Escherichia* and *Klebsiella* (Supplementary Fig. 9b). We hypothesize that this is due to gaps in reference genome collections as well as dissimilarity between strains of species that are common to reference collections and metagenomic data from this cohort.

**No strong signals of interaction between human DNA variation and microbiome content detected.** All participants in this study were recruited based on their participation in the first phase of the Africa Wits-INDEPTH partnership for Genomic Studies (AWI-Gen) study, which evaluated genomic and environmental risk factors for cardiometabolic disease in sub-Saharan African populations[56]. This study included human genome profiling of all participants using the Human Heredity and Health in Africa (H3Africa) single nucleotide polymorphism (SNP) array. While we have a very small sample size to assess interaction between human genetic variation and microbiome population, our study is one of the relatively few to characterize both human and microbiome DNA. Therefore, we performed association tests between key microbiome genera abundance levels and human SNPs. After correcting for multiple testing there were only a few human genomic SNPs with borderline statistically significant association with microbial genera abundance levels (Supplementary Table 3). These SNPs occur in genomic regions with no obvious connection to the gut microbiome (additional details in Supplementary Information). Additionally, we observe that microbiome samples do not cluster by self-reported ethnicity of the participant (Supplementary Fig. 10).

**South African gut microbiomes share taxa with western and nonwestern populations yet harbor distinct features.** To place the microbiome composition of South African individuals in global context with metagenomes from healthy adults living in other parts of the world, we compared publicly available data from five cohorts (Fig. 3a and Supplementary Table 4) comprising adult individuals living in the United States[1], northern Europe (Sweden)[57], agriculturalists living in Burkina Faso[28] and rural Madagascar[12], and the Hadza hunter-gatherers of Tanzania[7]. We grouped these datasets by lifestyle into the general categories of "nonwestern" (Tanzania, Madagascar, Burkina Faso), "western" (USA, Sweden), and South African (Bushbuckridge, Soweto). We note the caveat that these samples were collected at different times using different approaches, and that there is variation in DNA extraction, sequencing library preparation and sequencing, all of which may contribute to variation between studies. Recognizing this limitation, we observe that South African samples cluster between western and nonwestern populations in MDS (Fig. 3b) as expected, and that the first axis of MDS correlates well with geography and lifestyle (Fig. 3c). The relative abundance of Spirochaetaceae, Succinivibrionaceae, Bacteroidaceae, and Prevotellaceae are most strongly correlated with the first axis of MDS (Spearman's $\rho$ >0.75): Bacteroidaceae decreases with MDS 1 while Spirochaetaceae, Succinivibrionaceae, and Prevotellaceae increase (Fig. 3b). We observe a corresponding pattern of decreasing relative abundance of other VANISH taxa across lifestyle and geography (Supplementary Fig. 11). These observations suggest that the gut microbiome of South African cohorts is to some extent "intermediate" in composition when compared to cohorts at the extremes of western and nonwestern lifestyle.

The two South African cohorts also have distinct differences from both nonwestern and western populations, as evidenced by displacement along the second axis of MDS (Fig. 3b, c). To identify the taxa that drive this separation, we used DESeq2 to identify microbial genera that differed significantly in the South African cohort compared to both nonwestern and western categories (with the same directionality of effect in each comparison, e.g., enriched in South Africans compared to both western and nonwestern groups) (Supplementary Fig. 12). We observe that taxa including *Lactobacillus*, *Lactococcus*, and *Eggerthella* are lower in relative abundance in South Africans compared to both western and nonwestern groups. Conversely, *Klebsiella* and unclassified Christensenellaceae are enriched in South Africans.

**Within-species diversity across cohorts.** Having observed taxonomic differences at the species level between South Africans and other global populations, as well as between Soweto and Bushbuckridge, we hypothesized that strains of some species may differ between populations. We annotated the pangenome of the top six most abundant species on average across our cohorts and assessed whether pangenome content is significantly different

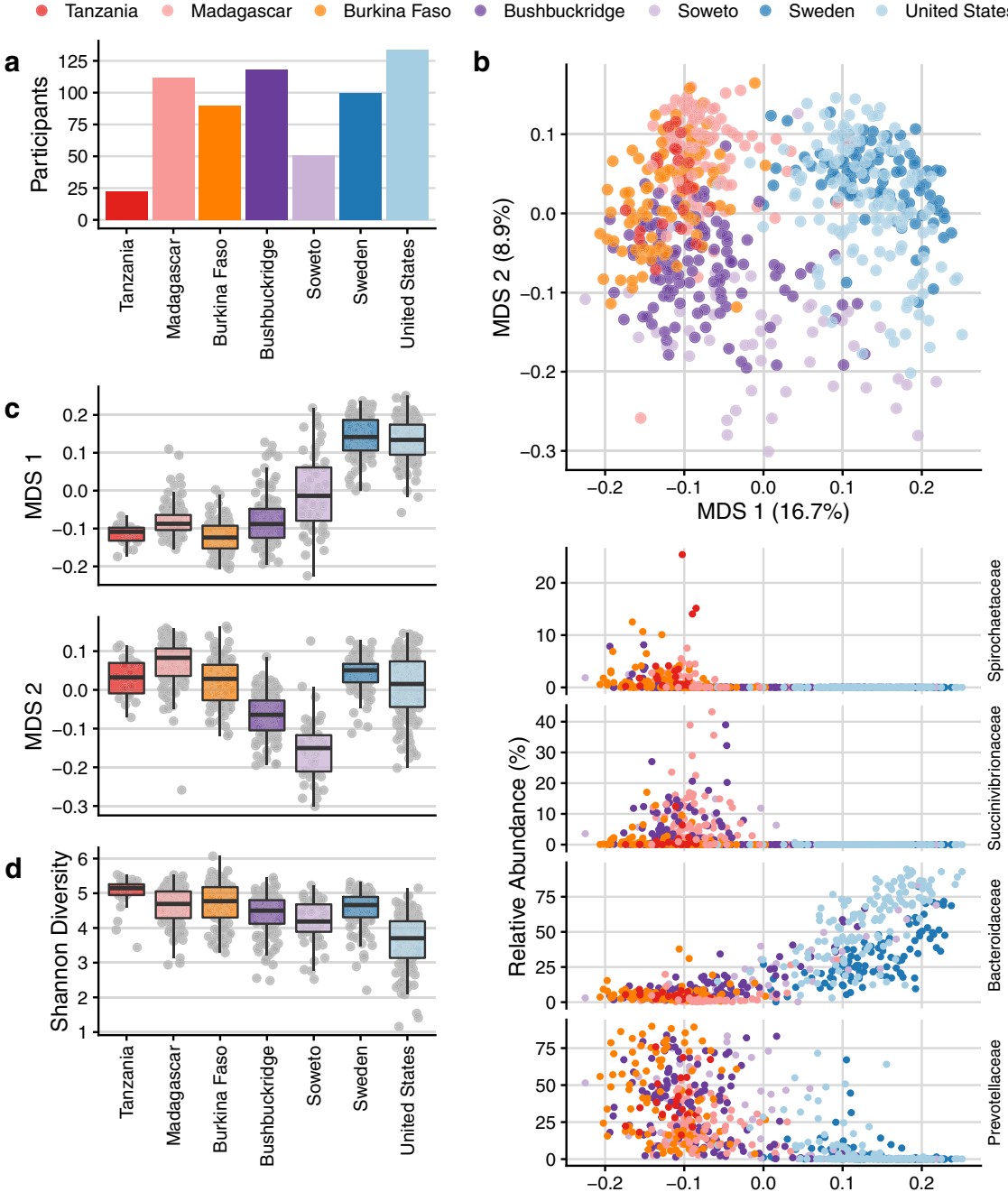

**Fig. 3 Community-level comparison of global microbiomes.** Comparisons of South African microbiome data to microbiome sequence data from four publicly available cohorts representing western (United States, Sweden) and nonwestern (Tanzania, Madagascar, Burkina Faso) populations. **a** Number of participants per cohort. **b** Multidimensional scaling of pairwise Bray–Curtis distance between samples from six datasets of healthy adult shotgun microbiome sequencing data. Western populations (Sweden, United States) cluster away from African populations practicing a traditional lifestyle (Madagascar, Tanzania, Burkina Faso) while transitional South African microbiomes overlap with both western and nonwestern populations. Shown below are scatterplots of relative abundance of the top four taxa most correlated with MDS 1 (Spearman's rho, Spirochaetaceae −0.824, Succinivibrionaceae −0.804, Bacteroidaceae 0.769, and Prevotellaceae −0.752) against multidimensional scaling axis 1 (MDS 1) on the *x*-axis. **c** Box plots of the first axis of MDS (MDS 1) which correlates with geography and lifestyle, and the second axis of MDS (MDS 2), which shows a distinct separation of South Africans from the other cohorts. **d** Shannon diversity across cohorts. Shannon diversity was calculated from data rarefied to the number of counts of the lowest sample. For box plots in **c** and **d**, lower and upper hinges correspond to the first and third quartiles, upper and lower box plot whiskers represent the highest and lowest values within 1.5 times the interquartile range, and the horizontal line represents the median. Participant sample size in **a–d** is as follows, with one sample per participant: *n* = 22 Tanzania, *n* = 112 Madagascar, *n* = 90 Burkina Faso, *n* = 118 Bushbuckridge, *n* = 51 Soweto, *n* = 100 Sweden, *n* = 134 United States.

between study sites (Supplementary Fig. 13). Interestingly, we find that *F. prausnitzii, B. vulgatus,* and *E. siraeum* indeed differ in pangenome content between Bushbuckridge and Soweto (PERMANOVA FDR-adjusted $q < 0.05$). *Prevotella copri* strains exhibit visible heterogeneity, but PERMANOVA is not significant after false discovery rate correction.

**Decreased sequence classifiability in nonwestern populations.** Given previous observations that gut microbiome alpha diversity is higher in individuals practicing traditional lifestyles[3,6,58] and that immigration from Southeast Asia to the United States is associated with a decrease in gut microbial alpha diversity[13], we hypothesized that alpha diversity would be higher in nonwestern populations, including South Africans, compared to western populations. We observe that Shannon diversity of the Tanzanian hunter-gatherer cohort is uniformly higher than all other populations (Fig. 3d; $q < 0.05$ for all pairwise comparisons; FDR-adjusted two-sided Wilcoxon rank-sum test) and that alpha diversity is lower in individuals living in the United States compared to all other cohorts (Fig. 3d; $q < 1.2e−05$ for all pairwise comparisons; FDR-adjusted two-sided Wilcoxon rank-sum test). Surprisingly, we observe comparable Shannon diversity between Madagascar and Sweden ($q > 0.05$, two-sided Wilcoxon rank-sum test). However, this could be an artifact of incomplete representation of diverse microbes in existing reference collections.

Existing reference collections are known to be limited in their ability to classify metagenomic sequences from nonwestern gut microbiomes[12,59], and we observe low sequence classifiability in nonwestern populations (Fig. 4a). Therefore, we sought orthogonal validation of our observation that South African microbiomes represent a transitional state between traditional and western microbiomes and employed a reference-independent method to evaluate the nucleotide composition of sequence data from each metagenome. We used the sourmash workflow[60] to compare nucleotide $k$-mer composition of metagenomic data and performed ordination based on angular distance, which accounts for $k$-mer abundance. Using a $k$-mer length of 31 ($k$-mer similarity at $k = 31$ correlates with species-level similarity[61]), we observe clustering reminiscent of the species ordination plot shown in Fig. 3, further supporting the hypothesis that South African microbiomes are transitional (Fig. 4b).

Previous studies have reported a pattern of higher alpha diversity but lower beta diversity in nonwestern populations compared to western populations[9,62]. Hypothesizing that alpha and beta diversity may be underestimated for populations whose gut microbes are not well-represented in reference collections, we compared beta diversity (distributions of within-cohort pairwise distances) calculated via species Bray–Curtis dissimilarity as well as nucleotide $k$-mer angular distance (Fig. 4c–e). Of note, beta diversity is highest in Soweto irrespective of distance measure, except for in a species-level comparison to the United States (Fig. 4c, FDR-adjusted Wilcoxon rank-sum test, $q < 5e−6$ for all tests). Intriguingly, in some cases we observe that the relationship of distributions of pairwise distance values changes depending on whether species or nucleotide $k$-mers are considered. For instance, considering only species content, Bushbuckridge has less beta diversity than Sweden, but this pattern is reversed when considering nucleotide $k$-mer content (Fig. 4d). Further, the same observation is true for the relationship between Madagascar and the United States (Fig. 4e) and Soweto and the United States. Additionally, we compared species and nucleotide beta diversity within each population using Jaccard distance, which is computed based on shared and distinct features irrespective of abundance. Considering nucleotide $k$-mers, all nonwestern populations have greater beta diversity than each western population

(Supplementary Fig. 14), though this is not the case when species annotations are considered. This indicates that gut microbiomes in these nonwestern cohorts have a longer "tail" of lowly abundant organisms that differ between individuals.

These observations are critically important to our understanding of beta diversity in the gut microbiome in western and nonwestern communities. In summary, we find evidence to challenge the existing dogma of an inverse relationship between alpha and beta diversity, and note that in some cases this existing generalization represents an artifact of limitations in reference databases used for sequence classification.

**Improving reference collections via metagenomic assembly.** Classification of metagenomic sequencing reads can be improved by assembling sequencing data into metagenomic contigs and grouping these contigs into draft genomes (binning), yielding metagenome-assembled genomes (MAGs). Notably, MAGs enable investigation of the genomes of uncultivatable organisms. While MAGs can suffer from incompleteness and contamination due to limitations of assembly and binning, software tools exist for evaluating MAG quality[63]. The majority of publications to date have focused on creating MAGs from short-read sequencing data[12,59,64], but generation of high-quality MAGs from long-read data from stool samples has recently been reported[65]. To better characterize the genomes present in our samples, we assembled and binned shotgun sequencing reads from South African samples into MAGs. We generated 2419 MAGs (39 high-quality, 2038 medium-quality, and 342 low-quality)[66] from 169 metagenomic samples (Supplementary Fig. 15a). Applying the criteria for near-complete genomes proposed by Nayfach et al. (≥90% completeness, ≤5% contamination, N50 ≥10 kb, average contig length ≥5 kb, ≤500 contigs, ≥90% of contigs with ≥5X read depth), 832 of these genomes (34%) are designated near-complete. Filtering for completeness greater than 75% and contamination less than 10% and de-replicating at 99% average nucleotide identity (ANI) yielded a set of 1342 nonredundant medium-quality or better representative strain genomes. This de-replicated collection includes VANISH taxa genomes, including 94 *Prevotella*, 41 *Prevotellamassilia*, 39 *Succinivibrio*, and 10 Spirochaetota (4 *Treponema_D*, 6 *UBA9732*) (Fig. 5a and Supplementary Data 7).

To assess this collection in the context of the known diversity of MAGs, we compared our de-replicated MAG set to the Unified Human Gastrointestinal Genome collection (UHGG)[67]. Of these 1342 representative strain genomes, 16 (1.2%) have <95% ANI to any genome in the full UHGG (Supplementary Fig. 15b) and 15 of these are retained in the final species set when de-replicated at 95% ANI against UHGG species representatives (Supplementary Data 7) (two genomes with less than 95% ANI to the UHGG species representatives were within 95% ANI of each other and thus only one was retained after dereplication). These 15 genomes represent 7 GTDB phyla (Supplementary Fig. 15c) and 13 of 15 genomes (87%) are from Bushbuckridge participants.

An additional 38 of 1342 genomes (2.8%) share at least 95% ANI compared to the UHGG species representatives, but are assigned a higher genome quality score by dRep than the corresponding UHGG representative (Supplementary Data 7, genome scoring metrics in "Methods" and ref. 100). We note that ANI is calculated on the basis of regions that align between genomes, and therefore may systematically underestimate genomic divergence in this genome collection.

Interestingly, many MAGs within this set represent organisms that are uncommon in western microbiomes or not easily culturable, including organisms from the genera *Treponema* and *Vibrio*. As short-read MAGs are typically fragmented and exclude

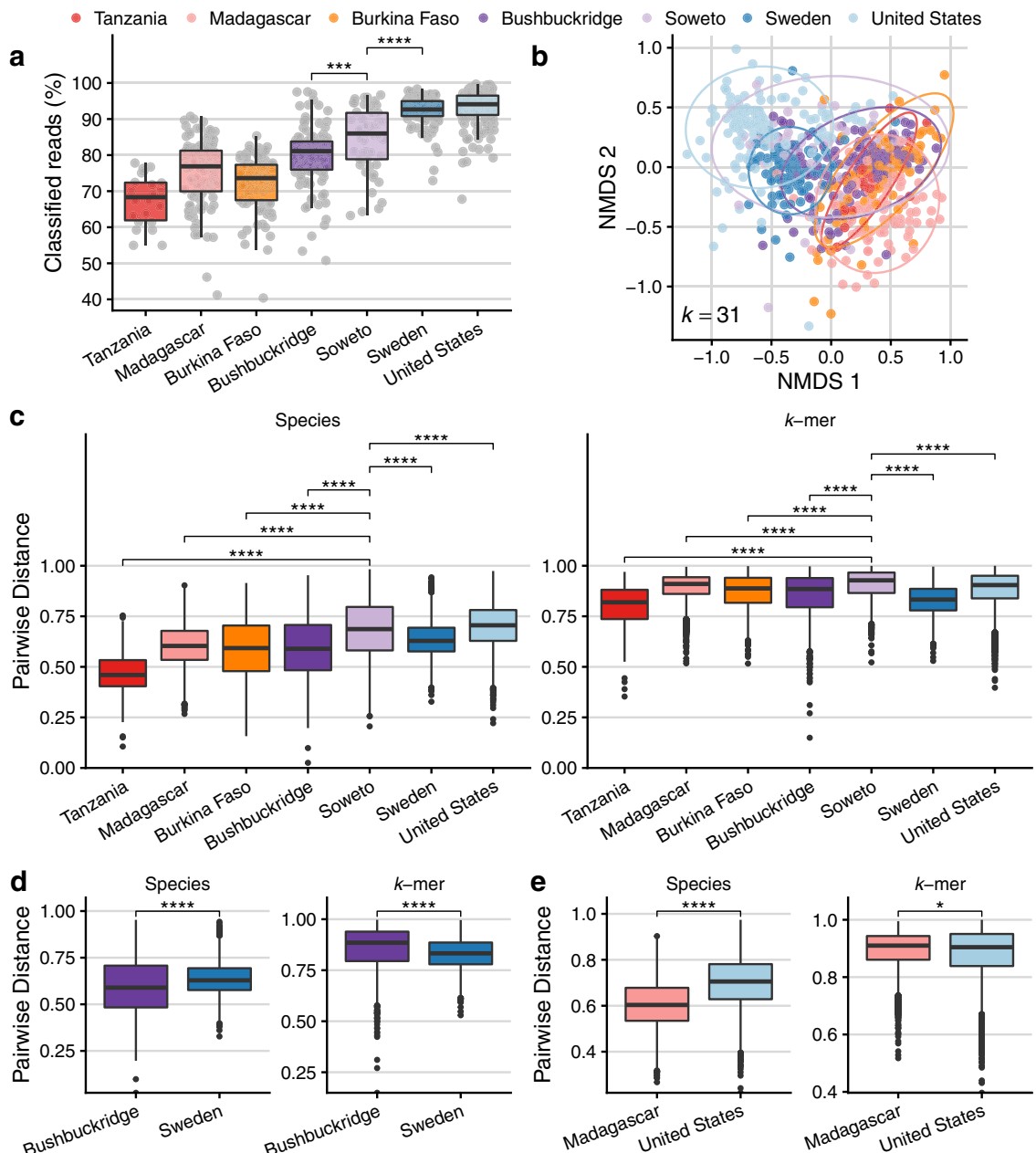

**Fig. 4 Comparison of beta diversity between communities calculated by taxonomy vs. nucleotide _k_-mer composition. a** Percentage of reads classifiable at any taxonomic rank, by cohort, based on a reference database of all genomes "scaffold" quality or higher in RefSeq and GenBank as of January 2020. Read classification is higher in western vs. nonwestern microbiomes (one-sided Wilcoxon rank-sum test between Soweto and Sweden, $p = 2.56e{-}8$), and higher in Soweto relative to Bushbuckridge (one-sided Wilcoxon rank-sum test, $p = 2.43e{-}4$). **b** Comparison of microbiome sequence data using _k_-mer sketches, a reference-free approach that allows comparison of nucleotide sequence composition. Briefly, a hash function generates signatures at varying sequence lengths ($k$) and _k_-mer sketches can be compared between samples. Plot shows non-metric multidimensional scaling (NMDS) of angular distance values between each pair of samples at $k = 31$ (approx. species-level)[61]. **c–e** Comparison of pairwise beta diversity within communities using Bray–Curtis distance for species and angular distance for nucleotide _k_-mer sketches. **c** Species beta diversity is higher in Soweto vs. all populations (one-sided Wilcoxon rank-sum test, FDR-adjusted $q < 2.7e{-}16$ for all tests) except for the United States, where beta diversity in Soweto is lower (one-sided Wilcoxon rank-sum test, $q = 4.05e{-}6$). Nucleotide _k_-mer diversity is higher in Soweto vs. all populations (one-sided Wilcoxon rank-sum test, FDR-adjusted $q < 2.2e{-}16$ for all tests). **d** Species beta diversity is higher in Sweden compared to Bushbuckridge, but nucleotide _k_-mer distance is higher in Bushbuckridge ($p < 2.22e{-}16$ for both tests). **e** Species beta diversity is higher in the United States cohort compared to the Malagasy, but nucleotide _k_-mer distance is higher in the Malagasy ($p < 2.22e{-}16$ species, $p = 0.034$ _k_-mer). For all box plots in **a, c–e**, lower and upper hinges correspond to the first and third quartiles, upper and lower box plot whiskers represent the highest and lowest values within 1.5 times the interquartile range, and the horizontal line represents the median. Significance values for two-sided Wilcoxon rank-sum tests denoted as follows: *$p < 0.05$, **$p < 0.01$, ***$p < 0.001$, ****$p < 0.0001$. One sample per participant, sample size in **a–e** is: $n = 22$ Tanzania, $n = 112$ Madagascar, $n = 90$ Burkina Faso, $n = 118$ Bushbuckridge, $n = 51$ Soweto, $n = 100$ Sweden, $n = 134$ United States.

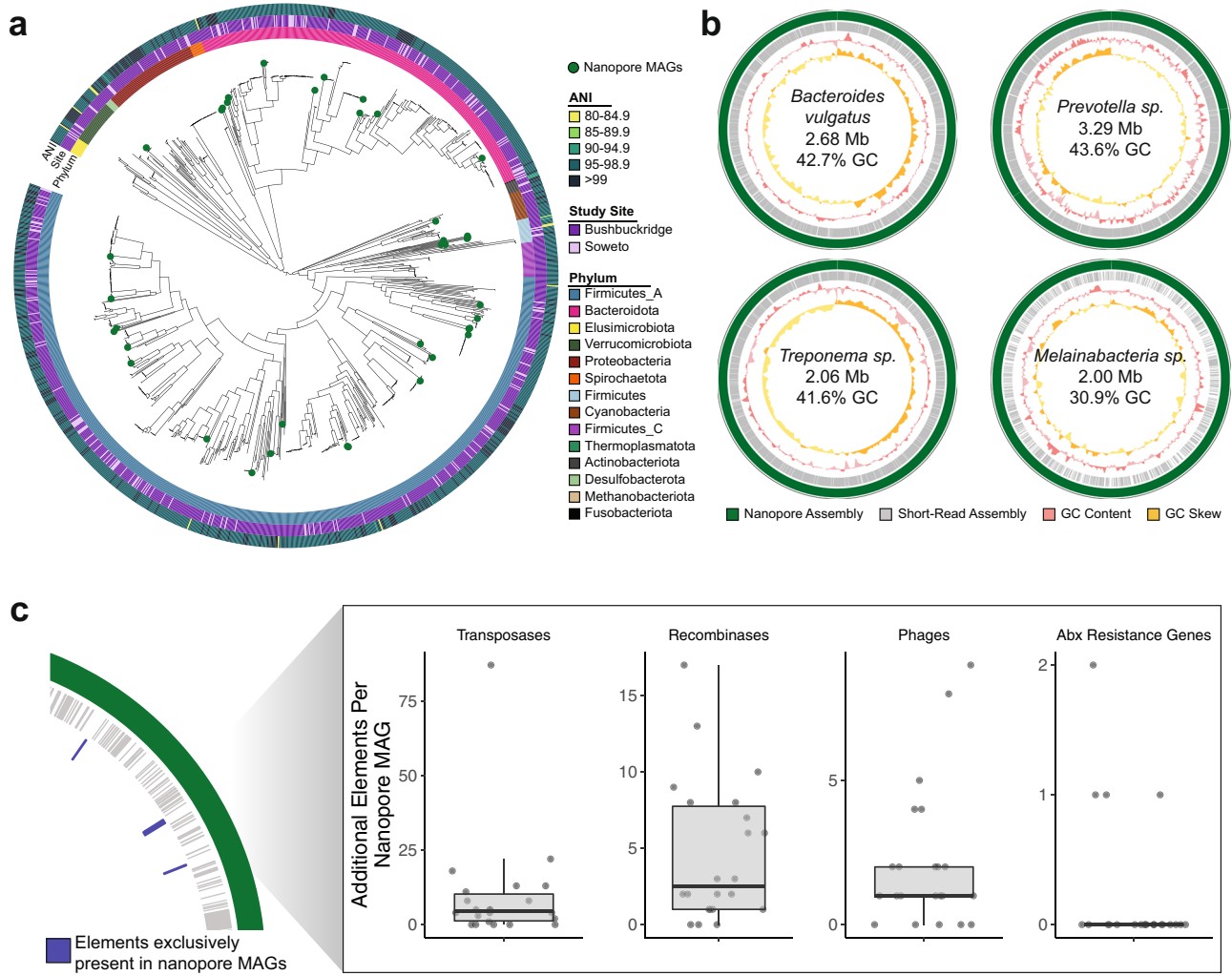

**Fig. 5 Complete and contiguous genomes of South African microbiota. a** Phylogenetic tree of de-replicated short-read MAGs and medium- and high-quality nanopore MAGs (green circles). Innermost ring indicates GTDB phylum, middle ring indicates study site associated with each MAG, and outer ring indicates the highest average nucleotide identity between each MAG and genomes from the UHGG. **b** A selection of MAGs assembled from long-read sequencing (green) of three South African samples compared contigs assembled from corresponding short-read data (gray). Third track (pink) indicates sliding genomic GC content, and fourth track (yellow) indicates sliding genomic GC skew. Breaks in circles represent different contigs. Genomic information within plots refer to assembly statistics of nanopore MAGs. **c** Number of additional genomic elements present in medium- and high-quality nanopore MAGs (*n* = 22) that are absent in corresponding short-read MAGs for the same organism, as diagrammed in the left-hand panel. Box plot lower and upper hinges correspond to the first and third quartiles, upper and lower box plot whiskers represent the highest and lowest values within 1.5 times the interquartile range, and the horizontal line represents the median. ANI average nucleotide identity, Mb megabase, Abx antibiotics, MAG metagenome-assembled genome.

mobile genetic elements, we explored methods to create more contiguous genomes, with a goal of trying to better understand these understudied taxa. We performed long-read sequencing on three samples from participants in Bushbuckridge with an Oxford Nanopore MinION sequencer (Supplementary Table 5, taxonomic composition of the three samples shown in Supplementary Fig. 16). Samples were chosen for nanopore sequencing on the basis of molecular weight distribution and total mass of DNA (see "Methods"). One flow cell per sample generated an average of 19.71 Gbp of sequencing with a read N50 of 8275 bp after basecalling. From our three samples, we generated 741 nanopore MAGs (nMAGs), which yielded 35 nonredundant genomes when filtered for completeness greater than 50% and contamination less than 10%, and de-replicated at 99% ANI (Supplementary Data 8, Fig. 5a, and Supplementary Fig. 17). Of these, 21 nMAGs (60%) assembled in a single contig (Table 2). All of the de-replicated nMAGs contain at least one full length 16S sequence, and the contig N50 of 28 of the 35 nMAGs (80%) is greater than 1 Mbp.

We compared assembly statistics between all MAGs and nMAGs, and find that while nMAGs are typically less complete when evaluated by CheckM, the contiguity of nanopore medium- and high-quality MAGs is an order of magnitude higher (mean nMAG N50 of 260.5 kb compared to mean N50 of medium- and high-quality MAGs of 15.1 kb) at comparable levels of average coverage (Supplementary Figs. 17 and 18). We expect that CheckM under-calculates the completeness of nMAGs due to the homopolymer errors common in nanopore sequencing, which result in frameshift errors when annotating genomes. Indeed, we observe that nMAGs with comparable high assembly size and low contamination to short-read MAGs are evaluated by CheckM as having lower completeness (Supplementary Fig. 18).

**Table 2 Nonredundant single-contig genomes assembled from nanopore sequencing.**

| Classification | Size (Mb) | Quality | 16S rRNAs | GC (%) | GC Skew |
|---|---|---|---|---|---|
| *Alistipes putredinis* | 1.91 | Medium | 2 | 53.1 | 0.96 |
| *Anaerotruncus* sp. | 2.04 | Medium | 2 | 43.71 | 0.94 |
| Bacilli bacterium | 1.46 | Medium | 1 | 26.19 | 0.93 |
| Bacteroidales bacterium | 2.79 | High | 4 | 49.82 | 0.92 |
| Bacteroidales bacterium | 1.7 | Medium | 1 | 56.6 | 0.7 |
| *Bacteroides vulgatus* | 2.68 | Medium | 3 | 42.71 | 0.84 |
| Candidatus Melainabacteria[a] | 2 | Medium | 1 | 30.9 | 0.32 |
| *Catabacter* sp.[a] | 1.65 | Medium | 1 | 46.4 | 0.87 |
| Clostridiales bacterium | 1.53 | Medium | 1 | 47.28 | 0.94 |
| Clostridiales bacterium | 2.65 | Medium | 3 | 42.82 | 0.69 |
| Clostridiales bacterium | 1.61 | Medium | 1 | 46.9 | 0.94 |
| *Clostridium* sp. | 1.53 | Medium | 1 | 25.24 | 0.89 |
| *Clostridium* sp. | 1.3 | Medium | 1 | 46.87 | 0.8 |
| *Clostridium* sp. | 2.01 | Medium | 3 | 28.81 | 0.92 |
| *Clostridium* sp. | 1.14 | Medium | 1 | 29.09 | 0.7 |
| Lentisphaeria bacterium[b] | 5.08 | Medium | 3 | 57.5 | 0.69 |
| Porphyromonadaceae bacterium | 2.97 | Medium | 5 | 47.43 | 0.76 |
| Ruminococcaceae bacterium | 2.27 | High | 3 | 51.43 | 0.91 |
| Ruminococcaceae bacterium | 1.78 | Medium | 3 | 58.25 | 0.63 |
| *Treponema* sp. | 2.06 | Medium | 3 | 41.55 | 0.93 |
| *Treponema succinifaciens* | 2.55 | High | 4 | 39.12 | 0.82 |
| uncultured *Ruminococcus* | 2.08 | Medium | 5 | 46.85 | 0.69 |

[a]nMAGs gained by long-read polishing.
[b]nMAGs improved by long-read polishing.

**Contiguous genomes generated through nanopore sequencing.** When comparing the de-replicated medium- and high-quality nMAGs with the corresponding short-read MAG for the same organism, we find that nMAGs typically include many mobile genetic elements and associated genes that are absent from the short-read MAG, such as transposases, recombinases, phages, and antibiotic resistance genes (Fig. 5c). Additionally, a number of the nMAGs represent highly contiguous genomes for their clades. For example, we assembled two single-contig, megabase-scale genomes from the genus *Treponema*, a clade that contains various commensal and pathogenic species (Fig. 5b). Notably, *Treponema* is a genus within the Spirochaetes phylum, which contains VANISH taxa and is often considered to be completely lost with industrialization[9,11]. While some *Treponema* species are known pathogens (*T. pallidum*), *Treponema* in non-industrialized communities is thought to serve as a mutualistic fiber degrader in response to different fiber-rich nonwestern diets[9]. The first of these genomes is a single-contig *Treponema succinifaciens* genome, classified as *Treponema_D succinifaciens* by GTDB. The type strain of *T. succinifaciens*, isolated from the swine gut[68], is the only genome of this species currently available in public reference collections. Our *T. succinifaciens* genome is an initial complete genome representative of this species from the gut of a human. We assembled a second *Treponema* sp. (GTDB *Treponema_D sp900541945*; Supplementary Fig. 19), which contains a candidate natural product biosynthetic gene cluster (aryl polyene cluster) and shares 92.1% ANI with *T. succinifaciens*. Additionally, we assembled a 5.08 Mbp genome for Lentisphaerae sp., an organism that has been shown to be significantly enriched in traditional populations[69]. This genome also contains an aryl polyene biosynthetic gene cluster and multiple beta-lactamases, and shares 94% 16S rRNA identity with *Victivallis vadensis* and is classified as *Victivallis sp900550905* by the GTDB, suggesting a previously undescribed species or genus of the family Victivallaceae and representing the second closed genome for the phylum Lentisphaerae.

Other nMAGs represent organisms that are prevalent in western individuals but challenging to assemble due to their genome structure. Despite the prevalence of *Bacteroides* in western microbiomes, only three closed *B. vulgatus* genomes are available in RefSeq. We assembled a single-contig, 2.68 Mbp *Bacteroides vulgatus* (GTDB *Parabacteroides sp900549585*) genome that is 65.0% complete and 2.7% contaminated and contains at least 16 putative insertion sequences, which may contribute to the lack of contiguous short-read assemblies for this species (Fig. 5b). Similarly, we assembled a single-contig genome for *Catabacter* sp., a species of the order *Clostridiales* (GTDB CAG-475 sp900550915 of the Christensenellales order); the most contiguous *Catabacter* genome in GenBank is in five scaffolded contigs[70]. The putative *Catabacter sp.* shares 85% ANI with the best match in GenBank, suggesting that it represents an undescribed species within the *Catabacter* genus or an undescribed genus, and it contains a sactipeptide biosynthetic gene cluster. Additionally, we assembled a 3.29 Mbp genome for *Prevotella* sp. (Fig. 5b, N50 = 1.14 Mbp), a highly variable genus that is prevalent in nonwestern microbiomes and associated with a range of effects on host health[71]. Notably, the first closed genomes of *P. copri*, a common species of *Prevotella*, were only recently assembled with nanopore sequencing of metagenomic samples; one from a human stool sample[65] and the other from cow rumen[72]. *P. copri* had previously evaded closed assembly from short-read sequence data due to the dozens of repetitive insertion sequences within its genome[65]. This *Prevotella* assembly contains cephalosporin and beta-lactam resistance genes, as well as an aryl polyene biosynthetic gene cluster.

Many long-read assembled genomes were evaluated to be of low completeness despite having contig N50 values greater than 1 Mbp. Analysis shows that many of these genomes had sparse or uneven short-read coverage, leading to gaps in short-read polishing that would otherwise correct small frameshift errors. To polish genomic regions that were not covered with short-reads, we performed long-read polishing on assembled contigs from each sample, and re-binned polished contigs. After long-read polishing, four additional nonredundant MAGs meet the criteria of greater than 50% completeness and less than 10% contamination, and 2 of the 35 MAGs generated using short-read

polishing improved in quality (Supplementary Data 8). Long-read polishing improved the completeness of many organisms that are not commonly described in the gut microbiota, due perhaps to their low relative abundance in the average human gut, or to biases in shotgun sequencing library preparation that limit their detection. For example, we generated a 2 Mbp Melainabacteria genome (GTDB species *UMGS1477 sp900552205* of the family Gastranaerophilaceae) (Fig. 5b). Melainabacteria is a non-photosynthetic phylum closely related to Cyanobacteria that has been previously described in the gut microbiome and is associated with consuming a vegetarian diet[47]. Melainabacteria have proven difficult to isolate and culture, and the only complete, single-scaffold genome existing in RefSeq was assembled from shotgun sequencing of a human fecal sample[47].

Interestingly, our Melainabacteria genome has a GC content of 30.9%, and along with assemblies of a *Mycoplasma sp.* (GTDB *CAG_460 sp000437315* of class Bacilli) (25.3% GC) and *Mollicutes sp.* (GTDB *Tener-01 sp001940985* of the class Bacilli) (28.1% GC) (Supplementary Fig. 20), represent AT-rich organisms that can be underrepresented in shotgun sequencing data due to the inherent GC bias of transposon insertion and amplification-based sequencing approaches[73] (Supplementary Figs. 21 and 22). Altogether, these three genomes increased in completeness by an average of 28.5% with long-read polishing to reach an overall average of 70.9% completeness. While these genomes meet the accepted standards to be considered medium-quality, it is possible that some or all of these highly contiguous, megabase-scale assemblies are complete or near-complete yet underestimated by CheckM, perhaps due to incomplete polishing.

Altogether, we find that de novo assembly approaches are capable of generating contiguous, high-quality assemblies for diverse gut microbes, offering potential for investigation into the previously unclassified genomic content in the microbiomes of nonwestern communities. In particular, nanopore sequencing produced contiguous genomes for organisms that are difficult to assemble due to repeat structures (*Prevotella* sp., *Bacteroides vulgatus*), as well as for organisms that are AT-rich (*Mollicutes* sp., *Melainabacteria* sp.). We observe that long reads capture a broader range of taxa both at the read and assembly levels when compared to short reads, and that short- and long-read polishing approaches yield medium-quality or greater draft genomes for these organisms. This illustrates the increased visibility that de novo assembly approaches lend to the study of the full array of organisms in the gut microbiome.

## Discussion

Together with Oduaran et al.[30], we provide a description of gut microbiome composition in Soweto and Bushbuckridge, South Africa, and to our knowledge, the first effort utilizing shotgun and nanopore sequencing in South Africa to describe the gut microbiome of adults. In doing so, we increase global representation in microbiome research and provide a baseline for future studies of disease association with the microbiome in South African populations, and in other transitional populations.

We find that gut microbiome composition differs demonstrably between the Bushbuckridge and Soweto cohorts, further highlighting the importance of studying diverse communities with differing lifestyle practices. Interestingly, even though gut microbiomes of individuals in Bushbuckridge and Soweto share many features, we do observe enrichment of hallmark taxa associated with westernization in Soweto. These include *Bacteroides* and *Bifidobacterium*, which have been previously associated with urban communities[3], consistent with Soweto's urban locale in the Johannesburg metropolitan area.

We also observe enrichment in relative abundance of crAssphage and crAss-like viruses in Soweto relative to Bushbuckridge, with relatively high prevalence in both cohorts yet lower abundance on average of crAssphage clades Alpha and Delta compared to several other populations. This furthers recent work which revealed that crAssphage is prevalent across many cohorts globally[49], but found relatively fewer crAssphage sequences on the African continent, presumably due to paucity of available shotgun metagenomic data. Just as shotgun metagenomic sequence data enables the study of viruses, it also enables us to assess the relative abundance of human DNA or damaged human cells in the stool. Surprisingly, we observe a high relative abundance of human DNA in the raw sequencing data. We find a statistically significantly higher relative abundance of human DNA in samples from Soweto compared to those from Bushbuckridge. Future research may help illuminate the potential reason for this finding, which may include a higher proportion of epithelium disruption by invasive bacteria or parasites in Soweto vs. Bushbuckridge, and in South Africa in general, compared to other geographic settings. Alternatively, this may also be attributable to a higher baseline of intestinal inflammation and fecal shedding of leukocytes. Without additional information, it is difficult to speculate on the reason for this finding.

We find that individuals in Bushbuckridge are enriched in VANISH taxa including *Succinatimonas*, which was recently reported to associate with microbiomes from individuals practicing traditional lifestyles[12]. Intriguingly, several VANISH taxa (*Succinatimonas*, *Succinivibrio*, *Treponema*) are bimodally distributed in the Bushbuckridge cohort. We hypothesize that this bimodality could be caused by differences in lifestyle and/or environmental factors including diet, history of hospitalization or exposure to medicines, physical properties of the household dwelling, or differential treatment of drinking water across the villages comprising Bushbuckridge. Additionally this pattern may be explained by participation in migration to and from urban centers (or sharing a household with a migratory worker). A higher proportion of men in the community engage in this pattern of rural-urban migration[39], but it is possible that sharing a household with a cyclical worker could influence gut microbiome composition via horizontal transmission[74].

Despite the fact that host genetics explain relatively little of the variation in microbiome composition[75], we do observe a small number of taxa that associate with host genetics in this population. Future work is required for replication and to determine whether these organisms are interacting with the host and whether they are associated with host health.

Additionally, we demonstrate marked differences between South African cohorts and other previously studied populations living on the African continent and western countries. Broadly, we find that South African microbiomes reflect the transitional nature of their communities in that they overlap with western and nonwestern populations. Tremendous human genetic diversity exists within Africa[76], and our work reveals that there is a great deal of unexplored microbiome diversity as well. In fact, we find that microbiome beta diversity within communities may be systematically underestimated by incomplete reference databases: taxa that are unique to individuals in nonwestern populations are not present in reference databases and therefore not included in beta diversity calculations. Though it has been reported that nonwestern and traditional populations tend to have higher alpha diversity but lower beta diversity compared to western populations, we show that this pattern is not universally upheld when reference-agnostic nucleotide comparisons are performed. By extension, we speculate that previous claims that beta diversity inversely correlates with alpha diversity may have been fundamentally limited by study design in some cases. Specifically, the

disparity between comparing small, homogenous African populations with large, heterogenous western ones constitutes a significant statistical confounder, potentially preventing a valid assessment of beta diversity between groups. Furthermore, alpha and beta diversity comparisons based on species-level taxonomic assignment may be further confounded due to the presence of polyphyletic clades in organisms like *Prevotella copri*[26,77], which are highly abundant in gut microbiomes of nonwestern individuals. Notably, we also demonstrate that the notion of a "western-nonwestern" axis of microbiome variation is over-simplified: we find taxa that are enriched in South Africans relative to both western and hunter-gatherer/agriculturalist cohorts.

Advances in sequencing technology enhance our ability to more thoroughly characterize microbiomes using culture-free approaches. Through a combination of short- and long-read sequencing, we successfully assembled contiguous, complete genomes for many organisms that are underrepresented in reference databases, including genomes that are commonly considered to be enriched in or limited to populations with traditional lifestyles including members of the VANISH taxa (e.g., *Treponema* sp., *Treponema succinifaciens*). The phylum Spirochaetes, namely its constituent genus *Treponema*, is considered to be a marker of traditional microbiomes and has not been detected in high abundance in human microbiomes outside of those communities[11,69]. Here, we identify Spirochaetes in the gut microbiome of individuals in urban Soweto, demonstrating that this taxon is not exclusive to traditional, rural populations, though we observe that relative abundance is higher on average in traditional populations. Generation of additional genomes of VANISH taxa and incorporation of these genomes into reference databases will allow for increased sensitivity to detect these organisms in metagenomic data. Additionally, these genomes facilitate comparative genomics of understudied gut microbes and allow for functional annotation of potentially biologically relevant functional pathways. We note that many of these genomes (e.g., Melainabacteria, *Succinatimonas*) are enriched in the gut microbiota of Bushbuckridge participants relative to Soweto, highlighting the impact of metagenomic assembly to better resolve genomes present in rural populations.

In addition to investigating members of the VANISH taxa, long-read sequencing enables the study of AT-rich genomes, which are difficult to sequence using transposon-based library construction approaches common in short-read studies. Thus, using long-read sequencing, we produced genomes for organisms that exist on the extremes of the GC content spectrum, such as *Mycoplasma* sp., *Mollicutes* sp., and *Melainabacteria sp.* We find that these organisms are sparsely covered by short-read sequencing, illustrating the increased range of non-amplification-based sequencing approaches, such as nanopore sequencing. Interestingly, these assemblies are evaluated as only medium-quality by CheckM despite having low measurements of contamination, as well as genome lengths and gene counts comparable to reference genomes from the same phylogenetic clade. We hypothesize that sparse short-read coverage leads to incomplete polishing and therefore retention of small frameshift errors, which are a known limitation of nanopore sequencing[78]. Further evaluation of 16S or long-read sequencing of traditional and western populations can identify whether these organisms are specific to certain lifestyles, or are more prevalent but poorly detected with shotgun sequencing.

While we find that the gut microbiome composition of the two South African cohorts described herein reflects their lifestyle transition, we acknowledge that these cohorts are not necessarily representative of all transitional communities in South Africa or other parts of the world which differ in lifestyle, diet, and resource access. Hence, further work remains to describe the gut microbiota in other understudied populations. This includes a detailed characterization of parasites present in microbiome sequence data, an analysis that we did not undertake in this study but would be of great interest. These organisms have been detected in the majority of household toilets in nearby KwaZulu-Natal province[79], and may interact with and influence microbiota composition[80].

Our study has several limitations. Although the publicly available sequence data from other global cohorts were generated with similar methodology to our study, it is possible that batch effects exist between datasets generated in different laboratories that may explain some percentage of the global variation we observe. Additionally, while nanopore sequencing is able to broaden our range of investigation, we illustrate that our ability to produce well-polished genomes at GC content extremes is limited. This may affect our ability to accurately call gene lengths and structures, although iterative long-read polishing improves our confidence in these assemblies. Future investigation of these communities using less biased, higher coverage short-read approaches or more accurate long-read sequencing approaches, such as PacBio circular consensus sequencing, may improve assembly qualities. Additionally, long-read sequencing of samples from a wider range of populations can identify whether the genomes identified herein are limited to traditional and transitional populations, or are more widespread. Further, future improvements in error rate of long-read sequencing may obviate the need for short-read polishing altogether.

Taken together, our results emphasize the importance of generating sequence data from diverse transitional populations to contextualize studies of health and disease in these individuals. To do so with maximum sensitivity and precision, reference genomes must be generated to classify sequencing reads from these metagenomes. Herein, we demonstrate the discrepancies in microbiome sequence classifiability across global populations and highlight the need for more comprehensive reference collections. Recent efforts have made tremendous progress in improving the ability to classify microbiome data through creating new genomes via metagenomic assembly[12,59,64], and here we demonstrate the application of short- and long-read metagenomic assembly techniques to create additional genome references. Our application of long-read sequencing technology to samples from South African individuals has demonstrated the ability to generate highly contiguous MAGs and shows immense potential to expand our reference collections and better describe microbiomes throughout diverse populations globally. In the future, microbiome studies may use a combination of short- and long-read sequencing to maximize information output, perhaps performing targeted nanopore or other long-read sequencing of samples that are likely to contain the most novelty on the basis of short-read data.

The present study was conducted in close collaboration between site staff and researchers in Bushbuckridge and Soweto as well as microbiome experts both in South Africa and the United States, and community member feedback was invited and incorporated at multiple phases in the planning and execution of the study (see Oduaran et al.[30] and Supplemental Information for additional detail). Tremendous research efforts have produced detailed demographic and health characterization of individuals living in both Bushbuckridge and Soweto[32,56,81,82] and it is our hope that microbiome data can be incorporated into this knowledge framework in future studies to uncover disease biomarkers or microbial associations with other health and lifestyle outcomes. More broadly, we feel that this is an example of a framework for conducting microbiome studies in an equitable manner, and we envision a system in which future studies of microbiome composition can be carried out to achieve detailed characterization of microbiomes globally while maximizing benefit to all participants and researchers involved.

## Methods

**Cohort selection.** Stool samples were collected from women aged 40–72 years in Soweto, South Africa and Bushbuckridge Municipality, South Africa. Participants were recruited on the basis of participation in AWI-Gen[56], a previous study in which genotype and extensive health and lifestyle survey data were collected. Human subjects research approval was obtained (Stanford IRB 43069, University of the Witwatersrand Human Research Ethics Committee M160121, Mpumalanga Provincial Health Research Committee MP_2017RP22_851) and informed consent was obtained from participants for all samples collected. Participants were not compensated for participation. Stool samples were collected and preserved in OmniGene Gut OMR-200 collection kits (DNA Genotek). Samples were frozen within 60 days of collection as per manufacturer's instructions, followed by long-term storage at −80 °C. As the enrollment criteria for our study included previous participation in a larger human genomics project[56], we had access to self-reported ethnicity for each participant (BaPedi, Ndebele, Sotho, Tsonga, Tswana, Venda, Xhosa, Zulu, Other, or Unknown). Samples from participants who tested HIV-positive or who did not consent to an HIV test were not analyzed.

**Metagenomic sequencing of stool samples.** DNA was extracted from stool samples using the QIAamp PowerFecal DNA Kit (QIAGEN Cat. No. 12830) according to the manufacturer's instructions except for the lysis step, in which samples were lysed using the TissueLyser LT (QIAGEN Cat. No. 85600) (30 s oscillations/3 min at 30 Hz). DNA concentration of all DNA samples was measured using Qubit Fluorometric Quantitation (DS DNA High-Sensitivity Kit, Thermo-Fisher Cat. No. Q32851). DNA sequencing libraries were prepared using the Nextera XT DNA Library Prep Kit (Illumina Cat. No. FC-131-1096). Final library concentration was measured using Qubit Fluorometric Quantitation and library size distributions were analyzed with the Bioanalyzer 2100 (Agilent G2939BA). Libraries were multiplexed and 150 bp paired-end reads were generated on the HiSeq 4000 platform (Illumina). Samples with greater than ~300 ng remaining mass and a peak fragment length of greater than 19,000 bp (with minimal mass under 4000 bp) as determined by a TapeStation 2200 (Agilent G2964AA) were selected for nanopore sequencing. Nanopore sequencing libraries were prepared using the 1D Genomic DNA by Ligation protocol (Oxford Nanopore Technologies SQK-LSK109) following standard instructions. Each library was sequenced with a full FLO-MIN106D R9 Version Rev D flow cell on a MinION sequencer for at least 60 h.

**Literature review.** Literature review criteria based on Brewster et al.[4] were employed: PubMed, EMBASE, SCOPUS, and Web of Science were queried for observational and interventional research involving the human gut microbiota through January 2021. Terms including "gut microbiome" and "gut microbiota" and names of each of the 54 African countries were included in the search. Primary reports on the gut microbiome in African children and/or adults, utilizing either 16S rRNA or shotgun metagenomic sequencing and written in English, were included. Abstracts, secondary reports, poster presentations, reviews or editorials, and in vivo and in vitro studies were excluded. The list of relevant articles yielded by this search strategy was manually reviewed.

### Computational methods

*Preprocessing and taxonomy profiling.* Stool metagenomic sequencing reads were trimmed using TrimGalore v0.6.5[83] with a minimum quality score of 30 for trimming (–q 30) and minimum read length of 60 (–length 60). Trimmed reads were deduplicated to remove PCR and optical duplicates using htstream Super-Deduper v1.2.0 with default parameters. Reads aligning to the human genome (hg19) were removed using BWA v0.7.17-r1188[84]. Taxonomy profiles were created with Kraken v2.0.9-beta with default parameters[85] and (1) a comprehensive custom reference database containing all bacterial and archaeal genomes in GenBank assembled to "complete genome," "chromosome," or "scaffold" quality as of January 2020, and (2) the pre-built Struo[50] GTDB release 95 database containing one genome per species. Bracken v2.2.0 was then used to re-estimate abundance at each taxonomic rank[86]. MetaPhlAn3[52] taxonomy profiles were also generated.

*Additional data.* Published data from additional adult populations were downloaded from the NCBI Sequence Read Archive or European Nucleotide Archive (Supplementary Table 4) and preprocessed and taxonomically classified as described above. The study by Backhed et al. sampled both mothers and infants: only the maternal samples were retained in this study. For datasets containing longitudinal samples from the same individual, one unique sample per individual was chosen (the first sample from each individual was chosen from the United States Human Microbiome Project cohort).

*K-mer sketches.* K-mer sketches were computed using sourmash v2.0.0[60]. Low abundance k-mers were trimmed using the "trim-low-abund.py" script from the khmer package v3.0.0[87] with a k-mer abundance cutoff of 3 (-C 3) and trimming coverage of 18 (-Z 18). Signatures were computed for each sample using the command "sourmash compute" with a compression ratio of 1000 (–scaled 1000)

and k-mer lengths of 21, 31, and 51 (-k 21,31,51). Two signatures were computed for each sample: one signature tracking k-mer abundance (–track-abundance flag) for angular distance comparisons, and one without this flag for Jaccard distance comparisons. Signatures at each length of k were compared using "sourmash compare" with default parameters and the correct length of k specified with the -k flag.

*Functional annotation.* Unassembled metagenomic reads were functionally profiled using ShortBRED[88] v0.9.3 with a pre-built antibiotic resistance database based on the Comprehensive Antibiotic Resistance Database[89]. Features were pre-filtered for >10% prevalence and statistical analysis was performed using MaAsLin v2[90] using the compound Poisson linear model (CPLM) and total sum scaling normalization with "site" as a fixed effect.

Pangenomes were calculated with PanPhlAn v3.1[52] using parameters for increased sensitivity recommended by the authors of the tool: "–min_coverage 1–left_max 1.70–right_min 0.30".

MetaCyc pathways were profiled with HUMAnN v3.0.0[52] with default parameters, using the mpa_v30_CHOCOPhlAn_201901 database. Forward and reverse reads were concatenated into one file per sample prior to processing. Pathway abundances were normalized to copies per million and statistical analysis was performed using MaAsLin v2 using the CPLM and total sum scaling normalization with "site" as a fixed effect.

*Genome assembly, binning, and evaluation.* Short-read metagenomic data were assembled with SPAdes v3.15[91] and binned into draft genomes using a publicly available workflow (https://github.com/bhattlab/bhattlab_workflows/blob/master/binning/bin_das_tool_manysamp.snakefile, commit version bbe6511 as of Apr 20, 2021). Briefly, short reads were aligned to assembled contigs with BWA v0.7.17[84] and contigs were subsequently binned into draft genomes with MetaBAT v2.15[92], CONCOCT v1.1.0[93], and MaxBin v2.2.7[94]. Default parameters were used for each binner, with the following exceptions: For the jgi_summarize_bam_contig_depths step of MetaBAT, minimum contig length was set at 1000 bp (–minContigLength 1000), minimum contig depth of coverage of 1 (–minContigDepth 1), and a minimum end-to-end percent identity of reads of 50 (–percentIdentity 50). Bins were aggregated and refined with DASTool v1.1.1[95]. Bins were evaluated for size, contiguity, completeness, and contamination with QUAST v5.0.2[96], CheckM v1.0.13[97], Prokka v1.14.6[98], Aragorn v1.2.38[99], and Barrnap v0.9 (https://github.com/tseemann/barrnap/). We referred to published guidelines to designate genome quality[66]. Individual contigs from all assemblies were assigned taxonomic classifications with Kraken v2.0.9[66,85]. To create de-replicated genome collections, genomes with completeness greater than 75% and contamination less than 10% (as evaluated by CheckM) were de-replicated using dRep v3.2.0[100] with ANI threshold to form secondary clusters (-sa) at 0.99 (strain-level) or 0.95 (species-level). For comparison to UHGG species representatives, secondary ANI was set to 0.95. dRep chooses the genome with the highest score as the cluster representative according to the following formula: dRep score $= A \times$ Completeness $- B \times$ Contamination $+ C \times$ (Contamination $\times$ (Strain heterogeneity/100)) $+ D \times$ log(N50) $+ E \times$ log(size) $+ F \times$ (centrality$-$secondary ani). $A$ through $F$ are values which can be tuned by the user to change the relative importance of each parameter in choosing representative genomes. Default parameters ($A = 1$, $B = 5$, $C = 1$, $D = 0.5$, $E = 0$, $F = 1$) were used herein.

Long-read data were assembled with Lathe v1[65]. Briefly, Lathe implements basecalling with Guppy v2.3.5, assembly with Flye v2.4.2[101], and short-read polishing with Pilon v1.23[102]. Contigs greater than 1000 bp were subsequently binned into draft genomes with MetaBAT v2.13 using minimum contig depth coverage of 1, minimum end-to-end percent identity of reads of 50, and otherwise using default parameters, then classified, and de-replicated as described above. Additional long-read polishing was performed using four iterations of polishing with Racon v1.4.10[103] and long-read alignment with minimap2 v2.17-r941[104], followed by one round of polishing with Medaka v0.11.5 (https://github.com/nanoporetech/medaka). Single-contig genomes were analyzed for GC skew using SkewIT v1[105]. Genomes of interest were plotted with the DNAPlotter GUI v18.1.0[106].

Draft genomes were additionally classified with GTDBtk v1.4.1 (classify_wf)[107] using release 95 reference data.

Direct comparisons between nMAGs and corresponding MAGs were performed by de-replicating high- and medium-quality nMAGs with MAGs assembled from the same sample. MAGs sharing at least 99% ANI with an nMAG were aligned to the nMAG regions using nucmer v3.1 and uncovered regions of the nMAG were annotated with prokka 1.14.6, VIBRANT v1.2.1[108], and ResFams v1.2[109].

Phylogenetic trees for all de-replicated short- and long-read MAGs were constructed with GTDBtk v1.4.1 and visualized with iTOL v6[110]. To construct phylogenetic trees for taxa of interest, reference 16S rRNA sequences were downloaded from the Ribosomal Database Project (Release 11, update 5, September 30, 2016)[111] and 16S rRNA sequences were identified from nanopore genome assemblies using Barrnap v0.9 (https://github.com/tseemann/barrnap/). Sequences were aligned with MUSCLE v3.8.1551[112] with default parameters. Maximum-likelihood phylogenetic trees were constructed from the alignments

with FastTree v2.1.10[112,113] with default settings (Jukes-Cantor + CAT model). Support values for branch splits were calculated using the Shimodaira-Hasegawa test with 1000 resamples (default). Trees were visualized with FigTree v1.4.4 (http://tree.bio.ed.ac.uk/software/figtree/).

*Statistical analysis and plotting.* Statistical analyses were performed using R v4.0.2[114] with packages MASS v7.3-53[115], stats v4.0.2114, ggsignif v0.6.0[116], and ggpubr v0.4.0[117]. Alpha and beta diversity were calculated using the vegan package v2.6.0[118]. Two-sided Wilcoxon rank-sum tests were used to compare alpha and beta diversity between cohorts. Count data were rarefied and normalized via cumulative sum scaling and log2 transformation[119] prior to MDS. Data separation in MDS was assessed via PERMANOVA (permutation test with pseudo F ratios) using the adonis function from the vegan package. Differential microbial features between individuals living in Soweto and Bushbuckridge were identified from unnormalized count data output from Kraken 2 classification and Bracken abundance re-estimation (filtered for 20% prevalence and at least 500 sequencing reads per sample) using DESeq2 with the formula "~site"[120]. Plots were generated in R using the following packages: cowplot v1.0.0[121], DESeq2 v1.28.0[120], genefilter v1.70.0[122], ggplot2 v3.3.2[123], ggpubr v0.4.0, ggrepel v0.8.2[124], ggsignif v0.6.0, gtools v3.8.2[125], harrietr v0.2.3[126], MASS v7.3-53, reshape2 v1.4.4[127], tidyverse v1.3.0[128], and vegan v2.6.0.

**Reporting summary**. Further information on research design is available in the Nature Research Reporting Summary linked to this article.

## Data availability

All shotgun sequence data and metagenome-assembled genome sequences generated by this study are deposited in the NCBI Sequence Read Archive under BioProject PRJNA678454. Participant-level metadata (age, BMI, blood pressure measurements, and concomitant medications) and human genetic data are deposited in the European Genome-phenome Archive (EGA) under Study ID EGAS00001002482 and dataset ID EGAD00001006581. The participant metadata are available under restricted access due to ethics requirements for the parent AWI-Gen study; access can be obtained by request from the Human Heredity and Health in Africa Data Access Committee (DBAC) at https://catalog.h3africa.org/. Requests submitted before or during the third week of the month will be reviewed at a DBAC meeting during the first two weeks of the subsequent month, and the DBAC will notify requestors within a week of the meeting. Source data for figures are available at GitHub and Zenodo (https://doi.org/10.5281/zenodo.5715685). Reference data used in this study are available as follows: the Comprehensive Antibiotic Resistance Database release 1.1.8 is available at https://card.mcmaster.ca/. Unified Human Gastrointestinal Genome collection data are available in the European Nucleotide Archive under study accession ERP116715. Genome Taxonomy Database release 95 is available at https://data.gtdb.ecogenomic.org/releases/.

## Code availability

R code for analysis and figure generation is available at GitHub and Zenodo (https://doi.org/10.5281/zenodo.5715685)[129]. Data analysis workflows referenced in "Methods" are available at https://github.com/bhattlab/bhattlab_workflows.

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

## Acknowledgements

We thank the participants in our study for taking part in this research. Additionally, we thank the Bushbuckridge Community Advisory Group for their thoughtful recommendations on study procedure. We thank Karen Andrade for her contributions in planning the 2019 Community Advisory Group workshop. We thank the INDEPTH consortium for their support of this project. We thank the numerous fieldworkers at the Soweto DPHRU and Agincourt HDSS who enrolled participants and collected data. In particular, we thank Melody Mabuza, the field worker at Agincourt HDSS who oversaw collection of Bushbuckridge participant enrollment, and Jackson Mabasa, who managed sample collection in Soweto. We thank Michèle Ramsay (AWI-Gen PI), Yusuf Ismail and Amanda Haye for their contributions to organizational and sample processing aspects of the project. We thank the Stanford Research Computing Center and Ben Siranosian for their contributions to computational infrastructure and support. This work was supported in part by a grant from the Stanford Center for Innovation in Global Health and by NIH grant P30 CA124435, which supports the following Stanford Cancer Institute Shared Resource: the Genetics Bioinformatics Service Center. A.S.B was supported by the Rosenkranz prize, a Sloan Foundation Fellowship, and by an R01AI148623 from the National Institute of Allergy and Infectious Diseases. F.B.T. was supported by the National Science Foundation Graduate Research Fellowship and the Stanford Computational, Evolutionary, and Human Genetics Pre-Doctoral Fellowship. D.M. was supported by the Stanford Graduate Fellowships in Science and Engineering program. O.H.O. was partially supported by a Fogarty Global Health Equity Scholar award (TW009338). A.N.W. is supported by the Fogarty International Centre, National Institutes of Health under award number K43TW010698. The work was further supported by the South African National Research Foundation (CPRR160421162721) and a seed grant from the African Partnership for Disease Control. The AWI-Gen project is supported by the National Human Genome Research Institute (U54HG006938) as part of the H3A Consortium. The MRC/Wits Rural Public Health and Health Transitions Research Unit and Agincourt Health and Socio-Demographic Surveillance System, a node of the South African Population Research Infrastructure Network (SAPRIN), is supported by the Department of Science and Innovation, the University of the Witwatersrand and the Medical Research Council, South Africa, and previously the Wellcome Trust, UK (grants 058893/Z/99/A; 069683/Z/02/Z; 085477/Z/08/Z; 085477/B/08/Z). This paper describes the views of the authors and does not necessarily represent the official views of the National Institutes of Health (USA).

## Author contributions

A.S.B. and S.H. conceived of study and secured funding. A.S.B., S.H., Z.L., R.T., X.G.O., F.W., A.N.W., R.G.W., K.K., S.T., S.A.N., and V.S. organized study logistics and coordinated participant enrollment and sample collection. V.S., M.R.H., O.H.O., F.B.T., and R.B. contributed to sample preparation and sequencing. F.B.T., D.M., and S.H. performed data analysis. A.S.B., S.H., F.B.T., and D.M. wrote and edited the manuscript.

## Competing interests

The authors declare no competing interests.

## Additional information

**H3Africa AWI-Gen Collaborative Centre**

Godfred Agongo, Marianne Alberts[12], Stuart Ali, Gershim Asiki, Vukosi Baloyi, Palwendé Romuald Boua, Jean-Tristan Brandenburg, Francisco Camiña Ceballos, Tinashe Chikowore, Solomon Choma, Ananyo Choudhury, Nigel Crowther, Cornelius Debpuur, Mwawi Gondwe, Scott Hazelhurst, Kathleen Kahn, Christopher Khayeka-Wandabwa, Isaac Kisiangani, Catherine Kyobutungi, Zané Lombard, Given Mashaba, Felistas Mashinya, Theo Mathema, Lisa Micklesfield, Shukri Mohamed, Busisiwe Mthembu, Freedom Mukomana, Engelbert Nonterah, Shane A. Norris, Ovokeraye Oduaran, Abraham R. Oduro, F. Xavier Gómez-Olivé, Michèle Ramsay, Osman Sankoh, Dhriti Sengupta, Natalie Smyth, Cassandra Soo, Himla Soodyall, Herman Sorgho, Yaniv Swiel, Ernest Tambo, Pauline Tindana, Halidou Tinto, Furahini Tluway, Stephen Tollman, Rhian Twine, Alisha Wade, Ryan Wagner, Henry Wandera, Chodziwadziwa Kabudula, Daniel Ohene-Kwofie & Floidy Wafawanaka

[12]Deceased: Marianne Alberts. A list of members and their affiliations appears in the Supplementary Information.

