## [Peer Review File · Nature Communications]

REVIEWER COMMENTS

Reviewer #1 (Remarks to the Author):

The present study expanded our understanding of the human intestinal microbiota diversity by exhibiting the gut microbial composition of a South African cohort. Besides, they present complete and contiguous reference genomes that will enable further studies of gut microbiota in nonwestern populations. Overall, the topic of the present study was commonplace, and the experimental design was primary. I raised my main concern for the manuscript improvement.

1-I appreciate the short- and long-read metagenomic sequencing for the intestinal microbiome analysis. But the sample size was still small. Also, how do we know if the sampling location is representative enough?

2-The authors mainly focused on the taxonomic level of the gut microbiome. How about the microbial functional genes and metabolic pathways?

3-If possible, I would suggest the author explore the gut microbial feature in the South African cohort at the genomic mutation level or evolutionary scale.

4-The results of the microbiome and human genetic association testing were very confusing.

5-The diet information and the lifestyle associated metadata for the cohort microbiome analysis were crucial, which may also provide evidence for the author's "intermediate microbiomes" claim. Unfortunately, the information is missing.

6-How to understand "find that these microbiomes are in some respects intermediate between those of individuals living in high-income countries and individuals living in rural agriculturalist and hunter-gatherer communities"? Do you mean microbial composition or microbial beta diversity points? But it was based on a visual distance. How about the Unifrac distance and Bray-Curtis distance?

Minor concern:

1- Line 40, two "and"

2- Line 850-852, typo?

Reviewer #2 (Remarks to the Author):

Comments for

Short- and long-read metagenomics of urban and rural South African gut microbiomes reveal a transitional composition and novel taxa

Tamburini et al. performed both short- and long-read based metagenomic sequencing of gut microbiomes collected from two groups in South African, each representing either urban or rural cohort. They analyzed the microbial community composition of these samples and compared them with that of published human gut studies and databases, and found that these South African cohorts have different gut microbiomes compared to those of high-income countries, which indicate an insufficient sampling for a comprehensive global human gut microbial reference database. The authors thus reconstructed high quality (sometimes complete) genomes for some less-known / less-described taxa. The manuscript was well organized and written, the datasets were carefully analyzed and generally appropriately shown in Tables and/or Figures.

Major comments:

The most concerned point of this manuscript is that the authors claimed in the Abstract that (line 47) “Our results suggest that South Africa’s transitional lifestyle and epidemiological conditions are reflected in gut microbiota compositions”, however throughout the manuscript they only mentioned “epidemiological” several times in the Introduction and once in the Results, and no specific analyses and main Tables/Figures were included to support and show this. If it was only a hypothesis or speculation, the authors may have to remove this sentence from their abstract, otherwise, some more robust analyses and results and discussion should be included.

Minor comments:

Abstract

Line 40. Deleted one of the two “and”.

Line 43-44. “within-cohort beta diversity patterns” and “reference-agnostic sequence comparison patterns” are very unclear in Abstract, more clear descriptions should be used given that some may only have access to the Abstract not the full article.

Results

Line 168-171. Some of the samples actually were only with small sizes of reads, could the one in Figure 1A with high abundance of Firmicutes among them?

Line 211. Please provide reference(s) for “although it is undergoing rapid epidemiological transition”.

Line 223. Please state clearly how this genera correlates to diet and lifestyle? Prevotella -> non-western? Bacteroides -> western?

Line 416-420. For the background of metagenome-assembled genomes, the one describing obtaining complete genomes from metagenomes (Chen, Lin-Xing, et al. "Accurate and complete genomes from metagenomes." *Genome research* 30.3 (2020): 315-333.) should be acknowledged.

Line 433-441. The addition of less-described taxa genomes to the reference database is among the most significant contributions of this study, thus a summary Figure indicating this should be included in the main text.

Line 492 for example, GC skew is a good sign for checking assembly error(s) in complete (and circular) genomes, the authors should perform this analysis for the complete genomes they generated for any potential errors, as they will be references that many other researchers will use for their studies.

Methods

Line 821. Please make it clear if “functionally profiles” for reads or assembled sequences.

Line 834-835. The basic parameters of binning should be included for clarification.

Line 850-852. The formula is not visible, please modify.

Line 857. The minimum length of contigs and other basic parameters in MetaBAT binning should be included. And please explain why two versions of MetaBAT were used, or is it a typo?

Line 872. Please state if the 16S sequences identified from the nanopore genomes have been corrected for sequencing errors using illumina reads or not?

Tables

Table 2. Could the authors make it more clear why “Short Read Only” is here for “genomes assembled from nanopore sequencing”? Does it mean only short reads were used for polishing? If yes, the genomes should be assembled (and polished) from both nanopore and Illumina reads as well.

And, the number of scaffolds in each MAG shown here should be included in the table as well.

Figures

Figure 1. One of the samples from Soweto contained very high abundant Firmicutes sequences similar to those from “environmental samples”, which is very uncommon given that it was with very low (if detected) abundance in any of the other samples shown in this figure. The author may have to check if this corresponding sample was contaminated somehow. It may be painful if they have to exclude this sample for the manuscript as many figures need remake, but it is totally necessary if it is a contaminated sample.

Line 973-982. How were the families selected to show here? As Bacteroides is among the most genera across the samples but not included here.

Line 986-988. Could the “greater dispersion of Soweto samples” be due to sequencing depth? Or the authors may also have removed that bias before performing the analyses? Same question for subfigure B.

Figure 3. Please correct me if I was wrong, it is not unclear what the community-level comparison analyses were based on? 16S rRNA genes or other markers?

Line 1026. scaffold -> scaffolds

Line 1044-1046. How did the authors explain the opposite patterns?

Line 1055-1057. Did the authors check by mapping the Illumina reads to those nMGAs with additional genomic elements to see if they could also be found there? If the subpopulation with the additional genomic elements has a lower relative abundance than the subpopulation without those elements, metagenomic assembly based on short reads will only generate the consensus sequence (contig/scaffold) without them. The authors may need to check some of the cases as examples to be included in the manuscript.

Line 1064-1067. Is the genomic information inside the circles for nMAGs? Please indicate.

Reviewer #3 (Remarks to the Author):

The study is the first to assess microbiome metagenomes from South African populations that are characterized as transitioning from high-income countries and individuals living more traditional rural lifestyles.

The dataset alone is impactful for the field, including a large collection of metagenomes, and genome reconstruction of bacterial species of special interest to those that study the impact of human lifestyle changes. The analyses are consistent with current standards.

Major: Relevant dataset from (<https://doi.org/10.1038/s41598-021-81257-w>) is missing from the analysis (NCBI under BioProjectID PRJNA690543). There is a paucity of metagenome data from Africa, as the authors have noted; include this dataset appears relevant. Moreover, the above citation adds strength to the authors argument regarding reference biases.

Minor: The manuscript would benefit from style and grammar editing that is beyond the scope of this review.

Minor: The interpretations of data are incremental, but meaningful, with a focus on current trends in the field, such as the impact of industrialization, genome reconstruction of taxa of interest, and the substantial reference bias.

REVIEWER COMMENTS

Reviewer #1 (Remarks to the Author):

The present study expanded our understanding of the human intestinal microbiota diversity by exhibiting the gut microbial composition of a South African cohort. Besides, they present complete and contiguous reference genomes that will enable further studies of gut microbiota in nonwestern populations. Overall, the topic of the present study was commonplace, and the experimental design was primary. I raised my main concern for the manuscript improvement.

We thank the reviewer for their review and constructive comments. In particular, we appreciate the reviewer's acknowledgement of the substantial contributions of this manuscript in exploring gut microbiome diversity in understudied populations, and in presenting complete bacterial genomes for important microbial taxa.

As outlined below, we have taken many of the reviewer's suggestions to enhance the analyses performed, and we believe this has substantially improved the manuscript.

1-I appreciate the short- and long-read metagenomic sequencing for the intestinal microbiome analysis. But the sample size was still small. Also, how do we know if the sampling location is representative enough?

We thank the reviewer for acknowledging the importance of using both short-read and long-read metagenomic sequencing to study the human gut microbiome. Notably, this is one of very few papers that has studied the human gut using nanopore sequencing to date, and the first application of nanopore sequencing to the human gut microbiome that we are aware of in the African continent. Importantly, these two sequencing methods are complementary and allow us to ask and answer different questions, with short-read sequencing enabling us to explore high-resolution taxonomic diversity and strain variation in these communities, and with long-read sequencing on a select number of samples enabling us to provide complete, contiguous reference genomes for key taxa.

With regards to the sample size for this study, the short-read metagenomics component of this study is, to our knowledge, the largest shotgun metagenomics study of adult African individuals to date. The nanopore sequencing was indeed conducted on a small number of samples (3), due to limitations in the total amount of DNA per sample that was available for sequencing. Therefore, we limit our analysis to the construction of contiguous microbial genomes and comparison of these genomes to those generated from short-read sequencing and those available in public reference databases, rather than drawing conclusions about population-level taxonomic diversity and strain variation from these data.

We appreciate the reviewer's concern about how representative the communities studied herein are of broader populations. We agree with the reviewer that one must be cautious in applying findings from one or a few populations to a broader group. In fact, we argue in lines 695-700

that these data cannot be considered representative of *all* transitional communities across South Africa, nor can they be considered representative of populations across the African continent. Additionally, in lines 720-737 we discuss the pressing need to more broadly study diverse global populations along the spectrum of industrialization to create more comprehensive reference databases and improve our ability to understand the relationship between the gut microbiome and human health. Africa is very diverse – environmentally, socially, economically, human genetics and care must be taken not to flatten a complex landscape. Rather than overgeneralizing the findings from the two communities studied in this manuscript, we choose to provide an in-depth characterization of the microbiomes of individuals in these communities, contextualize these findings against other datasets, and contribute new and valuable data (in the form of sequencing reads and metagenome-assembled genomes) that can be incorporated into reference databases. These are good examples of communities undergoing epidemiological transition.

2-The authors mainly focused on the taxonomic level of the gut microbiome. How about the microbial functional genes and metabolic pathways?

We thank the reviewer for the suggestion of incorporating functional analysis into this manuscript. We have done so as outlined in the fourth paragraph of this response to reviewers comments.

First, Supplementary Figure 8 investigates the antibiotic resistance profiles of each individual's gut metagenome, and we identify differentially abundant antibiotic resistance genes between individuals in rural Bushbuckridge and urban Soweto.

Second, regarding broader functional analysis, it is important to keep in mind that functional annotations are very biased toward genes found in well-studied organisms (e.g. *E. coli*) and that many open reading frames lack a known annotation in public reference databases, especially for organisms that are less prevalent in western/industrialized populations. This has been described in Jacobson et al. 2021, who describe limitations in high-resolution functional classification of shotgun sequencing reads from non-industrial populations (with only 25-30% of gene abundance classified to species level in hunter-gather and rural agriculturalist populations, as opposed to 65-75% of genes in industrial populations). Additionally, Pasolli et al. find a striking discrepancy in gene annotations across the taxonomy tree, with >90% of genes annotated in well-studied species and as few as 22% of genes annotated in poorly described genomes. Therefore, we anticipate that differential functional profiles between the communities described herein would be confounded by reference database limitations.

Third, recognizing this limitation, we have included MetaCyc pathway profiles generated by the Humann3 software for each metagenome in our South African cohort in Supplementary Figure 9. This analysis reveals several pathways which are differentially abundant between communities. We acknowledge in the text that these profiles may be biased toward well-studied organisms.

3-If possible, I would suggest the author explore the gut microbial feature in the South African cohort at the genomic mutation level or evolutionary scale.

We thank the reviewer for this suggestion. While we agree that it would be of interest to explore evolutionary relationships between gut metagenomic strains in this dataset, we feel that these analyses would be out of the scope of the current manuscript, which is already quite detailed and lengthy, and would require a dedicated followup study to fully define and test such hypotheses.

4-The results of the microbiome and human genetic association testing were very confusing.

We thank the reviewer for bringing this confusion to our attention. We have edited this section for clarity, including increased context behind the SNP profiling conducted on study participants. Additionally, the methods and additional findings are discussed in the Supplementary Information.

5-The diet information and the lifestyle associated metadata for the cohort microbiome analysis were crucial, which may also provide evidence for the author's "intermediate microbiomes" claim. Unfortunately, the information is missing.

We acknowledge and agree with the reviewer's desire for additional dietary and lifestyle information on this cohort. This is clearly of great interest, and as the reviewer knows, collecting these types of data is resource intensive. Thus, while collecting dietary data via food frequency questionnaire or other instrument was beyond the scope of the resources for the current project, the relationship between diet and microbiome composition in rural and urban South Africans is of intense interest for further study.

Regarding additional participant-level data, we would like to bring attention to metadata deposited for this study in EGA under accession EGAS00001002482, which contains additional anthropometric data about these participants, including blood pressure and rapid blood glucose measurements.

To add further context as to diet and lifestyle for participants living in Bushbuckridge and Soweto, we have extended the Supplementary Information to describe previous research on dietary practices and the relationship of diet and BMI for individuals living in these communities.

6-How to understand "find that these microbiomes are in some respects intermediate between those of individuals living in high-income countries and individuals living in rural agriculturalist and hunter-gatherer communities"? Do you mean microbial composition or microbial beta diversity points? But it was based on a visual distance. How about the Unifrac distance and Bray-Curtis distance?

We thank the reviewer for bringing up this point of confusion regarding which findings support the statement in the abstract. The statement in the abstract that we "find that these microbiomes

are in some respects intermediate between those of individuals living in high-income countries and individuals living in rural agriculturalist and hunter-gatherer communities” is summarizing the following findings:

- Figure 3: Multidimensional scaling of pairwise Bray-Curtis distance between samples demonstrates that the first axis of variation correlates with geography (3B, 3C), with South African samples intermediate between samples from Madagascar and Tanzania and samples from Sweden and the USA. The first axis of variation also correlates with abundance of hallmark “western” taxa, such as Bacteroidaceae” and abundance of hallmark “nonwestern” taxa, such as Spirochaetaceae and Prevotellaceae.
- Figure 4: The fraction of unclassified shotgun sequencing reads of South African samples are intermediate between that of samples from Madagascar and Tanzania and samples from Sweden and the USA. This result shows that South African samples are intermediate in the amount of unclassified microbial matter they contain.
- Supplementary Figure 11: South African samples are frequently intermediate in their abundance of VANISH taxa, having lower abundance than individuals in Madagascar and Tanzania, but higher abundance than individuals from the USA and Sweden.

In summary, we show that South African microbiomes are intermediate with regards to Bray-Curtis distance, abundance of taxa of interest (VANISH taxa, Prevotellaceae, Bacteroidaceae, etc.), as well as the fraction of reads that may represent unclassified, presumably novel taxa.

To clarify our metrics of microbiome similarity: we evaluated microbiome dissimilarity using Bray-Curtis dissimilarity when comparing the composition of samples from Soweto and Bushbuckridge in Fig. 2A, and find that samples from these two sites have distinct centroids (PERMANOVA $p < 0.001$). We also use Bray-Curtis dissimilarity when comparing these communities against global populations in Fig. 3B, and we evaluate within-population Bray-Curtis dissimilarity in Figures 4C, 4D, and 4E. We chose Bray-Curtis as our similarity metric, as UniFrac distance is typically limited to 16S rRNA analysis due to the need to reliably place taxa from each sample onto a phylogenetic tree to evaluate relatedness.

Minor concern:

1- Line 40, two “and”

We thank the reviewer for catching this typo.

2- Line 850-852, typo?

These lines refer to the formula that dRep implements to score genomes and identify cluster representatives. We have ensured that this formula appears correctly in the final document.

Reviewer #2 (Remarks to the Author):

Comments for Short- and long-read metagenomics of urban and rural South African gut microbiomes reveal a transitional composition and novel taxa

Tamburini et al. performed both short- and long-read based metagenomic sequencing of gut microbiomes collected from two groups in South African, each representing either urban or rural cohort. They analyzed the microbial community composition of these samples and compared them with that of published human gut studies and databases, and found that these South African cohorts have different gut microbiomes compared to those of high-income countries, which indicate an insufficient sampling for a comprehensive global human gut microbial reference database. The authors thus reconstructed high quality (sometimes complete) genomes for some less-known / less-described taxa. The manuscript was well organized and written, the datasets were carefully analyzed and generally appropriately shown in Tables and/or Figures.

We thank the reviewer for their kind appraisal of our work. In particular, we appreciate the reviewer highlighting the importance of global representation in building human gut reference databases, and our contribution of complete genomes for taxa that are of interest to the field yet have few or no references available.

Major comments:

The most concerned point of this manuscript is that the authors claimed in the Abstract that (line 47) “Our results suggest that South Africa’s transitional lifestyle and epidemiological conditions are reflected in gut microbiota compositions”, however throughout the manuscript they only mentioned “epidemiological” several times in the Introduction and once in the Results, and no specific analyses and main Tables/Figures were included to support and show this. If it was only a hypothesis or speculation, the authors may have to remove this sentence from their abstract, otherwise, some more robust analyses and results and discussion should be included.

We thank the reviewer for sharing this perspective, and we agree that additional analysis would be required to support the hypothesis that epidemiological conditions/transitions are reflected in gut microbiome composition. We have followed the author’s recommendation to remove this sentence from the abstract.

Minor comments:

Abstract

Line 40. Deleted one of the two “and”.

We thank the reviewer for catching this typo.

Line 43-44. “within-cohort beta diversity patterns” and “reference-agnostic sequence comparison patterns” are very unclear in Abstract, more clear descriptions should be used given that some may only have access to the Abstract not the full article.

We appreciate this feedback from the reviewer and have changed the sentence to instead state, “We demonstrate that reference collections are incomplete for characterization of the microbiomes of individuals living outside high-income countries, resulting in artificially low species-level beta diversity measurements.” We hope that this will be more clear to the reader.

Results

Line 168-171. Some of the samples actually were only with small sizes of reads, could the one in Figure 1A with high abundance of Firmicutes among them?

The sample in Fig 1A with a high abundance of Firmicutes (sample SWT9) was sequenced to a depth of 33.8M raw reads, with 23.6M human reads removed after de-duplication and 5.3M microbial reads remaining after all quality control. This sample is depicted in red in the figure below:

The sample with a high abundance of Firmicutes is sequenced to an intermediate level of depth compared to other samples (depicted in panel A) yet has a strikingly high percentage of human reads. Nonetheless, the >5M microbial sequence reads that remain after QC provide ample information for taxonomy and functional profiling: as few as 0.5M reads are required for robust taxonomic and functional profiling (Hillmann et al. 2018). This is consistent with evidence from rarefaction curves of samples in this dataset: when we filter our dataset to remove extremely lowly abundant features with counts < 100 reads that likely represent artifacts of kraken classification and rarefy each sample to 200,000 counts, we still observe that rarefaction curves plateau, indicating that the full species richness is being sampled in our dataset.

The reviewer will also note that read depth was greater on average in Bushbuckridge compared to Soweto as a result of how samples were pooled for sequencing. We have rarefied our data where appropriate to control for this, and have indicated in the Methods and figure legends when rarefied data are used.

Regarding the high abundance of Firmicutes in this sample (SWT9), we provide a detailed explanation in response to a later question by this reviewer. To summarize, these Firmicutes genomes were labeled as “environmental” as a result of a data deposition or curation error in NCBI, but originated from the human gut. We have updated Figure 1 to clarify this.

Line 211. Please provide reference(s) for “although it is undergoing rapid epidemiological transition”.

We thank the reviewer for this suggestion. We have added citations to two studies that examine epidemiological transition in Bushbuckridge. Houle et al., “The Unfolding Counter-Transition in Rural South Africa: Mortality and Cause of Death, 1994–2009”, describe cause-specific mortality in Bushbuckridge over the course of fifteen years, identifying changing patterns in deaths caused by HIV, TB, other communicable diseases, and noncommunicable disease. Bawah et al., “The Evolving Demographic and Health Transition in Four Low- and Middle-Income Countries: Evidence from Four Sites in the INDEPTH Network of Longitudinal Health and Demographic Surveillance Systems”, study epidemiological transition in sites in four countries, including Bushbuckridge, and describe increasing incidence of noncommunicable disease and consistently high rates of deaths attributable to HIV and TB.

Line 223. Please state clearly how this genera correlates to diet and lifestyle? Prevotella -> non-western? Bacteroides -> western?

We thank the reviewer for noting the vague language. Yes, the Bacteroides:Prevotella gradient has been described as increasing Bacteroides in western populations, and increasing Prevotella in nonwestern populations. We have clarified our language in the text.

Line 416-420. For the background of metagenome-assembled genomes, the one describing obtaining complete genomes from metagenomes (Chen, Lin-Xing, et al. "Accurate and complete genomes from metagenomes." *Genome research* 30.3 (2020): 315-333.) should be acknowledged.

We agree with the reviewer that this citation should be acknowledged and will provide a helpful resource for readers of this paper. We appreciate the reviewer's suggestion and have added this citation.

Line 433-441. The addition of less-described taxa genomes to the reference database is among the most significant contributions of this study, thus a summary Figure indicating this should be included in the main text.

We thank the reviewer for noting that the construction of genomes for novel taxa is a highly significant contribution of this study. In accordance with the reviewer's suggestion of representing this novelty in the main text, Figure 5A now summarizes the metagenome-assembled genomes of this study, including notation of nanopore genomes and summary of the average nucleotide identity between each MAG and the closest relative in the UHGG database. Additionally, we have retained Supplemental Figure 15, which includes summary classifications for all novel short-read MAGs generated within this study.

Line 492 for example, GC skew is a good sign for checking assembly error(s) in complete (and circular) genomes, the authors should perform this analysis for the complete genomes they generated for any potential errors, as they will be references that many other researchers will use for their studies.

We thank the reviewer for this suggestion. We have calculated the GC skew values for each of our medium- or high-quality single contig genomes presented in Table 2 using the method described in Lu et al. 2020 (SkewIT: The Skew Index Test for large-scale GC Skew analysis of bacterial genomes). This paper presents a Skew Index Test (SkewIT) that returns a Skew Index (SkewI) ranging from 0 to 1, where lower values indicate low/no genome-wide GC skew and higher values indicate higher genome-wide GC skew. These SkewI values are included in the main manuscript in Table 2. We compared each genome's SkewI value to the typical range described by the SkewIT paper for each genus (based on genomes of that genus in RefSeq). In the table below, we show single contig genomes classified at the genus level, along with the SkewI lower threshold for genomes of that genus. Genomes that were not classified to the genus level are not included.

Genus	SkewIT Index Lower Threshold	Our SkewI Values
Alistipes	< 0.55	0.96
Bacteroides	0.71	0.84

Clostridium	0.93	0.7-0.92
Ruminococcus	<0.4	0.69
Treponema	0.19	0.82-0.93

All of our genus-level genomes fall above the SkewIT thresholds, except for the four genomes classified by Kraken2 as Clostridium sp. Interestingly, these four genomes have somewhat discordant GTDB classifications. As these genomes may not closely represent any genomes that are present in RefSeq, it is possible that we do not have a good sense of what to expect in terms of SkewI values for genomes in these clades. We note that these genomes have GC skew index values ranging from 0.7 to 0.92, on the higher end of our GC skew index values. Additionally, their genome length GC skew shows clear and consistent shifts along the genome (below).

To gain a broader view of typical Skewl values for taxa of interest (those that classified at higher-than-genus levels or had somewhat low Skewl values), we pulled all Genbank genomes of “Complete” quality for taxa of interest and classified Skewl values. We find that the genomes presented within our manuscript have Skewl values (dotted lines) within the observed range for

genomes of the same classification in Genbank. Notably, we could not validate certain taxa (e.g. Lentisphaerae) due to the absence of any single contig genomes in Genbank.

For taxa such as Lentisphaerae, which has no single contig genomes publicly available, we aligned long reads back to the assembled genome and visually inspected inflection points of GC skew for possible misassemblies. In the case of the Lentisphaerae genome, the average coverage of long reads mapped back to the genome is 29.6, which is quite high, and there are no bases that have fewer than 2 supporting long reads.

We note that our long read assembly workflow is quite conservative in assembly parameters. Specifically, our pipeline detects misassemblies by identifying regions spanned by zero or one long reads and breaking the assembly at points that do not have support from multiple long reads.

Methods

Line 821. Please make it clear if “functionally profiles” for reads or assembled sequences.

We have updated this statement to reflect that unassembled metagenomic reads were functionally profiled.

Line 834-835. The basic parameters of binning should be included for clarification.

For the most part, default parameters for each binner were utilized. The text has been updated to reflect this, as well as to clarify the occasions where non-default parameters were used. We have also linked to the publicly available binning workflow created and used by our lab herein to perform the metagenomic binning for reference.

Line 850-852. The formula is not visible, please modify.

We thank the reviewer for calling this to our attention, this has been fixed in the current manuscript version.

Line 857. The minimum length of contigs and other basic parameters in MetaBAT binning should be included. And please explain why two versions of MetaBAT were used, or is it a typo?

MetaBat2 was run using a minimum contig length of 1000 bp, minimum contig depth of 1, minimum end-to-end percent identity of reads of 50, and otherwise using default parameters. This information has been added to the methods.

Two versions of MetaBAT2 were used (2.13 and 2.15) because binning of short-read contigs was updated at a later point, after the long read binning had already been performed. Only minor changes were made to MetaBAT2 in between those versions and we do not anticipate that this would significantly influence the output of our analysis, if at all.

Line 872. Please state if the 16S sequences identified from the nanopore genomes have been corrected for sequencing errors using illumina reads or not?

The nanopore genomes were polished using short reads or a combination of short and long reads (polishing method indicated in Table 2) prior to identification of 16S sequences.

Tables

Table 2. Could the authors make it more clear why “Short Read Only” is here for “genomes assembled from nanopore sequencing”? Does it mean only short reads were used for polishing? If yes, the genomes should be assembled (and polished) from both nanopore and Illumina reads as well.

We thank the reviewer for pointing out this area of confusion. Indeed, the column that the reviewer notes refers to the polishing steps used to correct errors in the nanopore assembly. “Short Read Only” refers to genomes that were binned from an assembly that had only been polished with short reads, while “Long Read” refers to genomes that were binned from assemblies that had been polished with long reads and short reads. We have clarified the labeling in the table.

We polished the assemblies from each sample with two approaches - short reads alone, and short reads and long reads. We compared the bins that were built from each polishing approach and identified corresponding bins across approaches, based on average nucleotide identity. We selected the higher quality bin from each comparison. As described in the text, we found that long read polishing appeared to only improve completeness and contamination in low GC content organisms, likely due to sparse short read coverage of these genomes due to GC bias in certain short-read sequencing methods. It is well-precedented to exclusively use short-read polishing in most circumstances, and previous work has identified that when short read coverage is even, the combination of short and long read polishing does not improve mismatch correction (Moss et al 2020 - Supplementary Note 1).

And, the number of scaffolds in each MAG shown here should be included in the table as well.

We thank the reviewer for this suggestion. This information is now included in Table 2.

Figures

Figure 1. One of the samples from Soweto contained very high abundant Firmicutes sequences similar to those from “environmental samples”, which is very uncommon given that it was with very low (if detected) abundance in any of the other samples shown in this figure. The author may have to check if this corresponding sample was contaminated somehow. It may be painful if they have to exclude this sample for the manuscript as many figures need remake, but it is totally necessary if it is a contaminated sample.

We thank the reviewer for raising this important point, as this would undoubtedly raise concerns for some readers as well. We examined the provenance of the genomes contributing to the “Firmicutes: environmental samples” classifications in this sample (SWT9) and found two GenBank genomes in our reference database that contributed to the majority of the “Firmicutes: environmental samples” sequence read classifications.

Species	NCBI taxid	NCBI accession	NCBI BioProject
Firmicutes bacterium CAG:345	1263020	GCA_000433315.1	PRJEB841
Firmicutes bacterium CAG:449	1263023	GCA_000432895.1	PRJEB885

Both genomes were produced by an early effort to leverage co-abundant genes (CAGs) for binning of metagenomic contigs into draft genomes (Nielsen et al. 2014). Notably, both genomes were assembled from human gut metagenomic data from the MetaHit project. It is unclear why these were deposited as “environmental samples” in NCBI, given that they derived from gut metagenomic data.

This finding is corroborated by taxonomy profiles generated using the Genome Taxonomy Database (GTDB). The GTDB taxonomy tree is based on core protein alignment of GenBank and RefSeq genomes and differs from the NCBI taxonomy tree. NCBI and GTDB lineages for these species are as follows:

NCBI accession	Unfiltered NCBI taxonomy	GTDB taxonomy
GCA_000433315.1	d__Bacteria; x__Terrabacteria group; p__Firmicutes; x__environmental samples; s__Firmicutes bacterium CAG:345	d__Bacteria; p__Firmicutes; c__Bacilli; o__RFN20; f__CAG-288; g__CAG-345; s__CAG-345 sp000433315
GCA_000432895.1	d__Bacteria; x__Terrabacteria group; p__Firmicutes; x__environmental samples;	d__Bacteria; p__Firmicutes; c__Bacilli; o__RFN20; f__CAG-449; g__CAG-449; s__CAG-449

	s__Firmicutes bacterium CAG:449	sp000432895
--	---------------------------------	-------------

The GTDB profiles for this sample indicate that species CAG-345 sp000433315 and CAG-449 sp000432895 comprise about 38% of the species-level counts for this sample.

In conclusion, we are confident that these genomes originated from the human gut, though it is an open question as to why this particular participant was enriched in these taxa compared to the rest of the cohort.

Line 973-982. How were the families selected to show here? As *Bacteroides* is among the most genera across the samples but not included here.

Figure 1B is intended to specifically highlight VANISH (Volatile and/or Associated Negatively with Industrialized Societies of Humans) taxa, a category of bacterial taxa that have been described as uncommon in western microbiomes including the families Prevotellaceae, Succinobivibrionaceae, Spirochaetaceae (Fragiadakis et al. 2018; Sonnenburg and Sonnenburg 2019). Interestingly, this analysis reveals that VANISH taxa are present in individuals living in industrialized Soweto. We believe that this observation adds nuance to our understanding of the global prevalence of VANISH taxa by demonstrating that these taxa are not restricted to the gut microbiome of individuals practicing traditional lifestyles.

Line 986-988. Could the “greater dispersion of Soweto samples” be due to sequencing depth? Or the authors may also have removed that bias before performing the analyses? Same question for subfigure B.

We thank the reviewer for this insightful question. We replicate this finding when we rarefy our species-level feature table to a uniform depth of 1.44 M counts. To obviate this point of concern for prospective readers, we have re-generated this figure using the rarefied feature table and have updated the relevant section of Methods.

Figure 3. Please correct me if I was wrong, it is not unclear what the community-level comparison analyses were based on? 16S rRNA genes or other markers?

We appreciate the reviewer pointing out this area of confusion. The community-level comparisons in Fig. 3 are based on taxonomic relative abundance data from the shotgun sequencing for each sample. The classification was performed using Kraken2 against a custom reference database containing all bacterial and archaeal genomes in GenBank of scaffold, chromosome, or complete genome quality. We orthogonally validated these classifications using GTDB and using MetaPhlan3 in Supplementary Figure 7.

Line 1026. scaffold -> scaffolds

We thank the reviewer for noting this. In NCBI, Assembly Levels fall under the following categories: Contig, Scaffold, Chromosome, Complete Genome. We have edited the text to clarify and now refer to “scaffold” quality in quotation marks so that it is more clear that it is a categorical designation.

Line 1044-1046. How did the authors explain the opposite patterns?

This is an excellent question and an important point that we try to make in this manuscript. We believe that the opposite patterns arise because reference collections are biased toward organisms that are prevalent in western cohorts, whereas k-mer-based surveys do not require a nucleotide sequence to match to any known species. This bias can exist both on the species level, when novel species are present in nonwestern individuals, and on the strain level, when strains of a known species differ across geography, and reference collections are biased towards strains of that species that are prevalent in the west. We hypothesize that due to this bias, an increased number of gut metagenomic sequence reads from western individuals match to reference genome databases compared to nonwestern individuals. This may lead to detection of an increased number of species in western individuals, resulting in increased Bray-Curtis distance (a measure of beta diversity) between pairs of individuals on average. We hypothesize that on average, a gut microbial species in a non-western individual is less likely to be present in a reference collection, especially species that are not highly prevalent. Those species would then be missed, and would not contribute to Bray-Curtis distance calculation, thereby artificially deflating the beta diversity values for that community. When we remove this reference bias by directly examining nucleotide k-mers -- irrespective of their taxonomic origin -- it appears that in actuality some non-western populations harbor greater beta diversity. We chose to highlight the situations in which beta diversity appears greater in a western cohort when species-level data are considered to challenge the trope in the published literature that “alpha diversity is greater in nonwestern populations but beta diversity is greater in western populations.”

Line 1055-1057. Did the authors check by mapping the Illumina reads to those nMGAs with additional genomic elements to see if they could also be found there? If the subpopulation with the additional genomic elements has a lower relative abundance than the subpopulation without those elements, metagenomic assembly based on short reads will only generate the consensus sequence (contig/scaffold) without them. The authors may need to check some of the cases as examples to be included in the manuscript.

We thank the reviewer for this thoughtful point. Indeed, generally speaking, if a subpopulation of a given taxon has additional genomic elements, those elements may be excluded from the assembly for that organism. This has also been demonstrated in nanopore assemblies as well (for example, when insertion sequences are present in a small subpopulation and those elements are excluded from an assembly). In these cases, however, we find that these additional genomic elements are present in nMGAs because long reads are necessary to span these repetitive elements and place them in genomic context. With short read assembly, these elements are assembled but often exist as single, small contigs, unable to be assembled into a

longer contig with their flanking genomic neighborhood and unable to be binned, due to differences in coverage and composition relative to the rest of the genome. This can be the case even when the entire population of a given species contains that repetitive element, simply because it breaks the assembly.

When we map the full set of shotgun sequencing reads to the *Treponema succinifaciens* genome, for example, we see that open reading frames annotated as transposases do indeed have short reads that map to them. However, while the *Treponema succinifaciens* nanopore genome has 15 detected transposases, the corresponding short-read MAG has only 2 transposases. For transposases that were not detected in the short-read MAG, we see that the short-read MAG contigs break at those points, indicating that assembly failed to build a contiguous sequence at those regions due to the repetitive nature of the transposases. We do not see instances of short read contigs that span, but do not include, the regions in the nMAG that have transposases (which would indicate a strain subpopulation that does not have the transposase element).

Pictured below: The *Treponema succinifaciens* nMAG, with the corresponding short read MAG aligned to it (grey), the unbinned contigs from the short read assembly (orange), and transposases (blue) aligned in the inner rings. Note that transposases typically appear in

regions that do not have corresponding short read MAG contigs, but occasionally have corresponding unbinned contigs.

Line 1064-1067. Is the genomic information inside the circles for nMAGs? Please indicate.

We thank the reviewer for pointing out this lack of clarity. Yes, the genomic information refers to the nMAGs, and this is now indicated in the figure legend.

Reviewer #3 (Remarks to the Author):

The study is the first to assess microbiome metagenomes from South African populations that are characterized as transitioning from high-income countries and individuals living more traditional rural lifestyles.

The dataset alone is impactful for the field, including a large collection of metagenomes, and genome reconstruction of bacterial species of special interest to those that study the impact of human lifestyle changes. The analyses are consistent with current standards.

We thank the reviewer for their positive appraisal of the novelty of our manuscript, and for highlighting the significant contributions of metagenomes and metagenome-assembled genomes to the field.

Major: Relevant dataset from (<https://doi.org/10.1038/s41598-021-81257-w>) is missing from the analysis (NCBI under BioProjectID PRJNA690543). There is a paucity of metagenome data from Africa, as the authors have noted; include this dataset appears relevant. Moreover, the above citation adds strength to the authors argument regarding reference biases.

We thank the reviewer for bringing this recent publication to our attention. As the reviewer has mentioned, this publication represents a significant contribution to the field through both a careful evaluation of reference biases and through analysis of functional resilience in global microbiomes. We have therefore added this dataset to our analyses in Figures 3 and 4, and Supplementary Figures 11, 12, and 14.

In summary, we find that on the population level, microbiomes from participants in the Burkina Faso cohort overlap with Tanzania/Madagascar and also with South Africans (Figure 3). Consistent with data from Madagascar, Tanzania, and South Africa, metagenomes in the Burkina Faso cohort are enriched in VANISH taxa relative to westerners, with a particularly striking enrichment in *Treponema* spp. (Figure S11). As the reviewer noted, this dataset strengthens our argument that reference databases are incomplete for classifying nonwestern metagenomes, as the distribution of read classification rates in the Burkina Faso cohort is strikingly less than for western cohorts, and less even than the rural Malagasy cohort (Figure 4).

Minor: The manuscript would benefit from style and grammar editing that is beyond the scope of this review.

We have reviewed the manuscript thoroughly and edited for grammar, style, and clarity.

Minor: The interpretations of data are incremental, but meaningful, with a focus on current trends in the field, such as the impact of industrialization, genome reconstruction of taxa of interest, and the substantial reference bias.

We thank the reviewer for noting the important contributions and relevance of this manuscript.

References

- Fragiadakis, Gabriela K., Samuel A. Smits, Erica D. Sonnenburg, William Van Treuren, Gregor Reid, Rob Knight, Alphaxard Manjurano, et al. 2018. "Links between Environment, Diet, and the Hunter-Gatherer Microbiome." *Gut Microbes* 10 (2): 216–27.
- Hillmann, Benjamin, Gabriel A. Al-Ghalith, Robin R. Shields-Cutler, Qiyun Zhu, Daryl M. Gohl, Kenneth B. Beckman, Rob Knight, and Dan Knights. 2018. "Evaluating the Information Content of Shallow Shotgun Metagenomics." *mSystems* 3 (6).
<https://doi.org/10.1128/mSystems.00069-18>.
- Nielsen, H. Bjørn, Mathieu Almeida, Agnieszka Sierakowska Juncker, Simon Rasmussen, Junhua Li, Shinichi Sunagawa, Damian R. Plichta, et al. 2014. "Identification and Assembly of Genomes and Genetic Elements in Complex Metagenomic Samples without Using Reference Genomes." *Nature Biotechnology* 32 (8): 822–28.
- Sonnenburg, Erica D., and Justin L. Sonnenburg. 2019. "The Ancestral and Industrialized Gut Microbiota and Implications for Human Health." *Nature Reviews. Microbiology* 17 (6): 383–90.

REVIEWERS' COMMENTS

Reviewer #1 (Remarks to the Author):

The authors have addressed my concerns appropriately.

Reviewer #2 (Remarks to the Author):

I thank the authors for their careful responses to the comments that I raised on the original manuscript. Their modifications have resolved all my concerns.

Reviewer #3 (Remarks to the Author):

Our understanding of the human gut microbiome has well noted compositional difference between high-income derived lifestyles over the more traditional lifestyles, yet there are limited data on the process of this transformation. The authors demonstrate more transitional patterns in South Africa, which may be key to understanding the ecological process.

The authors well address all of my concerns, and more so; they did an impressive job with reviewer comments in general. The analysis and interpretations are clear and very well presented. The narrative is well edited. Overall, this work is a strong contribution and strong fit for Nature Communications.

Reviewer #1 (Remarks to the Author):

The authors have addressed my concerns appropriately.

We thank the reviewer for their comments and suggestions throughout the review process.

Reviewer #2 (Remarks to the Author):

I thank the authors for their careful responses to the comments that I raised on the original manuscript. Their modifications have resolved all my concerns.

We thank the reviewer for their suggestions and are glad that we have satisfactorily resolved all concerns.

Reviewer #3 (Remarks to the Author):

Our understanding of the human gut microbiome has well noted compositional difference between high-income derived lifestyles over the more traditional lifestyles, yet there are limited data on the process of this transformation. The authors demonstrate more transitional patterns in South Africa, which may be key to understanding the ecological process.

The authors well address all of my concerns, and more so; they did an impressive job with reviewer comments in general. The analysis and interpretations are clear and very well presented. The narrative is well edited. Overall, this work is a strong contribution and strong fit for Nature Communications.

We thank the reviewer for their thoughtful comments throughout the review process and for their kind words about our manuscript and our responses to previous rounds of review.